# LLaVA-CMoE: Towards Continual Mixture of Experts for Large Vision-Language Models

## Abstract

Mixture of Experts (MoE) architectures have recently advanced the scalability and adaptability of Large Language Models (LLMs) for continual multimodal learning. However, extending these models to accommodate sequential tasks remains challenging. As new tasks arrive, naive model expansion leads to rapid parameter growth, while modifying shared routing components often causes catastrophic forgetting, undermining previously learned knowledge. To address these issues, we propose **LLaVA-CMoE**, a continual learning framework for LLMs that requires no replay data of previous tasks and ensures both parameter efficiency and robust knowledge retention. Our approach introduces a Probe-Guided Knowledge Extension mechanism, which uses probe experts to dynamically determine when and where new experts should be added, enabling adaptive and minimal parameter expansion tailored to task complexity. Furthermore, we present a Probabilistic Task Locator that assigns each task a dedicated, lightweight router. To handle the practical issue that task labels are unknown during inference, we leverage a VAE-based reconstruction strategy to identify the most suitable router by matching input distributions, allowing automatic and accurate expert allocation. This design mitigates routing conflicts and catastrophic forgetting, enabling robust continual learning without explicit task labels. Extensive experiments on the CoIN benchmark, covering eight diverse VQA tasks, demonstrate that LLaVA-CMoE delivers strong continual learning performance with a compact model size, significantly reducing forgetting and parameter overhead compared to prior methods. These results showcase the effectiveness and scalability of our approach for parameter-efficient continual learning in large language models. Our code will be open-sourced soon.

## 1 Introduction

Multimodal Large Language Models (MLLMs) (Caffagni et al., 2024; Wu et al., 2023; Li et al., 2023; Radford et al., 2021; Liu et al., 2023) often employ Parameter-Efficient Fine-Tuning (PEFT) methods (Abou Baker et al., 2024; Han et al., 2024; Hu et al., 2022; Houlsby et al., 2019; Liu et al., 2022; 2021; Edalati et al., 2022; Zhang et al., 2022; Chen et al., 2022) after large-scale pre-training to efficiently adapt to downstream tasks without full retraining, ensuring computational efficiency and competitive performance. These training processes are commonly conceptualized as a multi-task learning (MTL) paradigm, where all tasks' data are simultaneously accessible. However, in real-world scenarios, knowledge and tasks are continuously updated in a streaming manner, posing new challenges for the efficient adaptation of MLLMs. This dynamic environment necessitates that fine-tuning methods not only remain parameter-efficient, but also support robust continual learning (CL) to enable models to incrementally acquire new knowledge from new tasks.

Catastrophic forgetting (Kirkpatrick et al., 2017; Chen & Liu, 2018), a fundamental issue in CL, also poses a significant challenge for achieving efficient continual learning in MLLMs. To address this issue, some approaches (Cai et al., 2023; Lei et al., 2023; Yang et al., 2023) involve storing data from previous tasks as subsets or distributions, and utilize a data replay scheme during new tasks training to prevent knowledge forgetting. However, as tasks multiply, the storage and computational overhead associated with replay-based methods can become prohibitive. Another line of works (Lao et al., 2023; Zheng et al., 2023; Farajtabar et al., 2020; Zhu et al., 2021; Chiaro et al., 2020; Serrà et al., 2018; Jha et al., 2024; Cai et al., 2022; He et al., 2023; Peng et al., 2021; Yang et al., 2023) focuses on designing novel loss functions or model architectures to alleviate forgetting. Nonetheless,

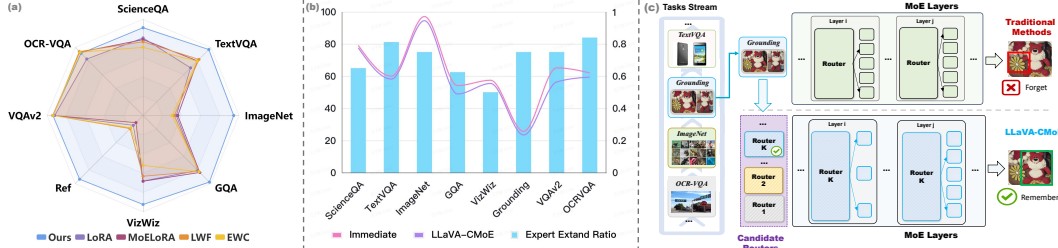

Figure 1: **a) Visualization of the Model's Anti-Forgetting Capability.** Our method significantly improves anti-forgetting capability, achieving a performance boost compared to baselines. **b) Model Forgetting Ratio vs. Parameter Expansion Rate.** *Immediate* means the performance just after training. After training on task streams (from left to right), our methods nearly match the performance of *Immediate* while saving nearly 30% parameters, enhancing efficiency. **c) Conceptual comparison between the previous method and our LLaVA-CMoE.** Unlike traditional methods that use a fixed number of experts, our model dynamically adjusts experts per layer based on task needs. Besides, we also build a router bank to reduce forgetting and improve knowledge retention. For clarity, we illustrate only the routing and experts within the block.

these approaches typically involve updating parameters shared across tasks, which may still lead to performance degradation on previously learned tasks due to interference.

Recently, Wang et al. (Wang & Li, 2024) propose to use a Mixture of Experts (MoE) architecture, adding new experts at all layers to acquire task-specific knowledge in continual learning. Though MoE's scalability makes it promising for continual learning, two key challenges remain: 1) *When and where to insert experts?* Existing work (Yang et al., 2024) shows that additional parameter demands vary greatly across tasks, indicating expert allocation should be adaptive to task similarity instead of statically predefined. (Zhong et al., 2024) explore dynamic allocation by analyzing expert selection distribution changes before/after task training, but such shifts only reflect task preference (not real need for new experts), leaving current methods struggling to accurately identify when/where new experts are required. 2) *How to mitigate catastrophic forgetting in the Router?* In MoE, the Router is crucial for dynamic expert allocation. Optimizing experts for new tasks may cause forgetting of prior knowledge, and fine-tuning the Router can alter earlier tasks' routing strategies, also causing catastrophic forgetting. (Yu et al., 2024) address this by fixing expert count and adding routers continuously to preserve prior routers' knowledge integrity. However, as tasks increase, a fixed expert set may limit the model's capacity to accommodate highly different tasks.

Building on the aforementioned challenges, we propose an effective framework, **LLaVA-CMoE**, enabling effective continual learning for MoE-based MLLM without requiring replay data. To address the challenge of adaptive expert expansion, we propose the **Probe-Guided Knowledge Extension (PKGE)** algorithm. This approach utilizes minimal probe data from new tasks to monitor the behavior of newly added expert modules at each layer. By analyzing per-layer probe activation frequencies, the model adaptively decides whether to expand capacity at that layer, preventing unnecessary parameter growth (Zhong et al., 2024) while preserving previously acquired knowledge during the learning of new tasks. To mitigate catastrophic forgetting arising specifically from *router updates*, we introduce the **Probabilistic Task Locator (PTL)**. Rather than relying on a single shared router that risks entangling task-specific routing preferences, PTL assigns each task a lightweight, dedicated router that preserves prior routing behavior. Activating a task's router reinstates its routing policy and task-specific performance with minimal overhead, effectively isolating tasks while maintaining scalability. During inference, when task identities are unknown, we adopt a reconstruction-based strategy inspired by variational methods (An & Cho, 2015; Pu et al., 2016; Pinheiro Cinelli et al., 2021). Using a VAE-based framework, the model learns task-specific distributions and infers task identities by evaluating reconstruction errors, enabling PTL to dynamically route inputs and maintain robust performance on unseen tasks.

We evaluate LLaVA-CMoE on the CoIN dataset (Chen et al., 2024), a benchmark tailored to assess continual learning performance across eight distinct VQA-based tasks. Through extensive quantitative and qualitative evaluations, along with comprehensive ablation studies, we demonstrate that our approach not only significantly mitigates catastrophic forgetting, but also promotes effective knowledge transfer and adaptation as new tasks are learned sequentially.

## 2 RELATED WORK

**Continual Learning.** Continual learning aims to mitigate catastrophic forgetting (Hassabis et al., 2017; Wu et al., 2024) and enable incremental knowledge acquisition. Existing methods fall into four main categories: 1) *Regularization-based* approaches (Lao et al., 2023; Zheng et al., 2023; Farajtabar et al., 2020; Zhu et al., 2021), which constrain updates to important parameters; 2) *Architecture-based* approaches (Chiaro et al., 2020; Serrà et al., 2018; Jha et al., 2024; Cai et al., 2022; Sun et al., 2021; He et al., 2023; Peng et al., 2021; Yang et al., 2023; Yu et al., 2024; Chen et al., 2023a), which add task-specific components to reduce interference; 3) *Replay-based* approaches (Cai et al., 2023; Lei et al., 2023; Yang et al., 2023; Lopez-Paz & Ranzato, 2017), which store or generate samples for rehearsal; and 4) *Prompt-based* methods (Wang et al., 2023b; Qian et al., 2023; D'Alessandro et al., 2023; Zheng et al., 2024), which use learnable prompts to maintain performance. However, computational and storage burdens, as well as forgetting due to parameter updates, remain open challenges (Chen et al., 2024).

**Mixture of Experts.** The MoE architecture (Riquelme et al., 2021; Shen et al., 2023; Mustafa et al., 2022) employs specialized expert networks and a gating mechanism for efficient computation. Sparsely-gated MoE (Shazeer et al., 2017; Lepikhin et al., 2020) has shown strong performance in LLMs, such as Mixtral 8x7B (Jiang et al., 2024), across diverse NLP tasks. Recent work (Liu et al., 2024; Yang et al., 2024; Chen et al., 2023b; Luo et al., 2024) combines MoE with LoRA (Hu et al., 2022) for more efficient training. MoE's scalability has led to its adoption in continual learning, e.g., LEMoE (Wang & Li, 2024) adds experts at all layers for new tasks, and Lifelong-MoE (Chen et al., 2023a) trains new experts while freezing old ones. CoIN (Chen et al., 2024) further explores MoELoRA, but existing methods still face parameter overhead and suboptimal robustness.

**Large Language Models & PEFT.** Recently, Large Language Models (Liu et al., 2023; Touvron et al., 2023; Chowdhery et al., 2023; Brown et al., 2020) have garnered widespread attention for their remarkable abilities in areas such as language generation, in-context learning, and reasoning. To enable data- and compute-efficient adaptation for specific downstream tasks, various PEFT methods (Karimi Mahabadi et al., 2021; Houlsby et al., 2019; Li & Liang, 2021) have been introduced. Among these, LoRA (Hu et al., 2022) stands out by representing weight updates through low-rank decomposition, keeping the original weights frozen while training only the new update matrices. In this study, we combined LoRA with MoE for efficient continual MLLM fine-tuning.

## 3 METHODS

In this section, we first clarify the problem formulation of multimodal continual learning in Section 3.1. We then elaborate on the proposed **LLaVA-CMoE** and its core components: PGKE and PTL in Section 3.2. Finally, we detail the training objectives in Section 3.3.

### 3.1 PROBLEM FORMULATION

Let $\{\mathcal{T}_1, \ldots, \mathcal{T}_N\}$ be a set of $N$ tasks, where each task $\mathcal{T}_i$ has its training set $\mathbf{D}^i$ consisting of $n_i$ multimodal inputs $\mathbf{X} \in \{\mathbf{X}_t^i, \mathbf{X}_{img}^i, \mathbf{X}_l^i\}_{i=1}^{n_i}$. Here, $\mathbf{X}_t^i$, $\mathbf{X}_{img}^i$, and $\mathbf{X}_l^i$ represent the textual input (instruction), image, and the corresponding answer, respectively. In Continual Learning, the model is trained sequentially on the $N$ tasks, and while training the $i$-th task $\mathcal{T}_i$, the model needs to maximize the probability $P_i$ through Next Token Prediction. Notably, while training the $i$-th task, it cannot access to the data of previous tasks $\{\mathcal{T}_1, \ldots, \mathcal{T}_{i-1}\}$. During the inference phase, we receive only task-agnostic data from either seen or unseen tasks. This practical constraint differentiates our approach from traditional task-incremental learning (Task-IL) paradigms, where explicit task identifiers are typically accessible during inference (Shi et al., 2024), thus significantly enhancing practical applicability in real-world scenarios.

### 3.2 LLAVA-CMOE

In this paper, we propose **LLaVA-CMoE**, a framework that enables efficient expert expansion while preserving strong anti-forgetting capabilities, as illustrated in Figure 2. First, in Section 3.2.1, we introduce the core architecture of the Mixture-of-Experts (MoE) framework. Next, we conduct an in-depth analysis of the forgetting problem in MoE-based architectures from two key components:

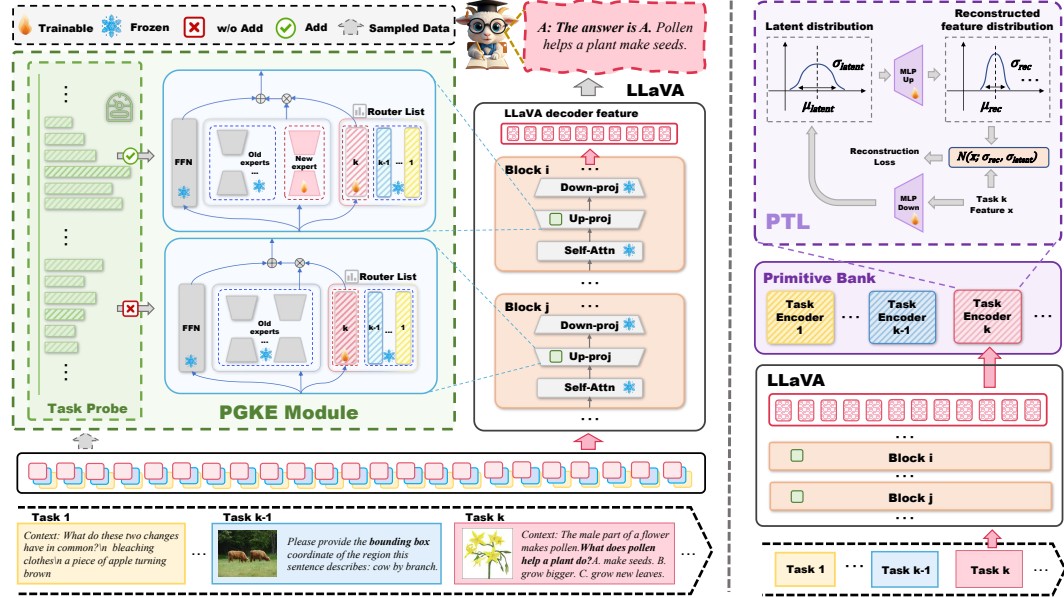

Figure 2: **Overview of our LLaVA-CMoE.** Our model consists of two main components: 1) **Probe-Guided Knowledge Expansion (PGKE)** adaptively expands experts for different tasks based on task probe guidance, enabling efficient task learning. 2) **Probabilistic Task Locator (PTL)** establishes the connection between task distributions and task routing. During inference, it identifies the corresponding router based on the input, ensuring accurate task-specific processing. As illustrated in the left part, Block $i$ was selected by PGKE to be expanded with new experts, while Block $j$ was not. The right side shows that the model stores the features of task $k$ into the primitive bank via PTL.

the experts and the router. For the experts, Section 3.2.2 presents the PGKE mechanism, which dynamically identifies optimal locations for expert expansion that against forgetting. For the router, Section 3.2.3 describes the PTL mechanism, which enables the model to effectively route inputs to task-relevant experts during inference, thereby ensuring comprehensive knowledge utilization.

### 3.2.1 MIXTURE OF EXPERT LAYER

Our model is built upon a multimodal large language model (MLLM), such as LLaVA (Liu et al., 2023), in which the MoE is implemented by augmenting the Feed-Forward Network (FFN) modules within each Transformer block. Furthermore, each expert is constructed using a LoRA module, parameterized by $\{\mathbf{A}_i, \mathbf{B}_i\}_{i=1}^{N_e}$ (Hu et al., 2022), where $N_e$ denotes the number of experts. In the $h$-th block, given the multimodal token $\mathbf{X}^h$, the output token $\mathbf{X}_{out}^h$ is computed as follows:

$$\mathbf{X}_{out}^h = \mathbf{W}_0\mathbf{X}^h + \sum_{i=1}^{N_e} \omega_i' \mathbf{B}_i\mathbf{A}_i\mathbf{X}^h, \quad \omega_i' = \frac{\omega_i}{\sum_{j=1}^{N_e} \omega_j}, \quad \omega_i = \exp(\mathbf{G}\mathbf{X}^h)\cdot\mathbb{I}[i \in \text{topk}(\mathbf{G}\mathbf{X}^h)], \quad (1)$$

where $\mathbf{W}_0$, $\mathbf{G}$ represent the linear layers' weights of the Feed-Forward Network and the router network, respectively. $\mathbb{I}$ denotes the indicator function.

### 3.2.2 PROBE-GUIDED KNOWLEDGE EXTENSION (PGKE)

Current MoE-oriented continual learning methods usually append a *fixed* number of experts to *every* layer (Wang & Li, 2024; Yang et al., 2024) when training a new task. However, this simple solution (i) wastes parameters when the new task is similar to previous ones, (ii) scales quadratically with the number of tasks, and (iii) still alters the shared router, inducing forgetting. To address these issues, we propose Probe-Guided Knowledge Extension (PGKE), **measuring *where* the current model lacks capacity *before* it allocates new experts**. The key intuition is that, if the features required by the incoming task already lie in the span of existing experts, then a freshly initialized "probe" expert will rarely be selected. Conversely, persistent high probe activations indicate that *no existing expert can explain the new examples*, indicating that extra capacity is truly needed. This "try-before-you-buy" principle keeps the parameter budget tight and aligns expansion with genuine knowledge gaps.

Following this principle, we design a two-stage training framework, consisting of *probe locating* and *expert expansion*. Given the training set $\mathbf{X}^i$ for the $i$-th task, we begin by sampling two non-overlapped subsets: $\mathbf{X}^i_{\text{train}}$ for training and $\mathbf{X}^i_{\text{eval}}$ for evaluation of the probe experts.

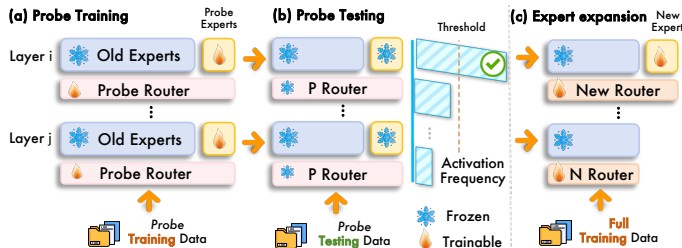

Figure 3: Illustration of Probe-Guided Knowledge Extension (PGKE) process. a) Probe experts are added to all layers. We use probe training data to train probe experts and probe routers. b) Calculating activation frequencies to select layers to expand. c) Expanding selected layers' experts and doing full training.

**Probe locating.** In this phase, each MoE layer is augmented by introducing new probe experts along with a corresponding probe router, denoted as $\mathbf{R}_i$. To ensure accurate probing, $\mathbf{R}_i$ is initialized with the parameters of the router from the $(i-1)$-th task, new probe experts are initialized with average weight of old expert group. An additional dedicated sub-router is appended to accommodate the probe experts. During this phase, the parameters of all existing experts are frozen; only the parameters of the probe experts and $\mathbf{R}_i$ are updated using $\mathbf{X}^i_{\text{train}}$.

Upon completion of probe training, we evaluate the activation statistics of all experts in each layer over the validation set $\mathbf{X}^i_{\text{eval}}$ by computing both the mean and variance of their activation frequencies (logit values). Based on these statistics, we define a threshold for expert expansion as follows (with the layer index omitted for clarity):

$$\text{Threshold} = \text{mean}(\mathbf{Act}) - \alpha \cdot \text{std}(\mathbf{Act}), \qquad (2)$$

where $\alpha$ is a hyperparameter that modulates the sensitivity of expert growth (set to 0.8). If the activation frequencies ($\mathbf{Act}$) of $N_s$ probe experts exceed this threshold, we interpret this as evidence that the current layer requires additional capacity, and therefore augment the layer by introducing $N_s$ new experts accordingly. Otherwise, we keep the current layer unchanged. The selection process is illustrated in Figure 3.

**Expert expansion.** After determining the optimal number of experts to be integrated into each layer, we perform expert expansion by augmenting the selected layers with the corresponding number of newly initialized experts and routers, following the same initialization strategy as for $\mathbf{R}_i$. Additionally, the weights of the new expert are derived from the weights of the expert that is most frequently activated during the probe locating process. Subsequently, the model is fine-tuned on the entire training dataset $\mathbf{X}^i$, freezing old experts to mitigate forgetting. To accommodate the varying complexities of different tasks, the parameter $N_s$, representing the number of newly added experts, is treated as a dynamically adjustable quantity. This approach not only enhances the efficiency of parameter expansion, but also ensures that the model can effectively adapt to and learn from tasks with greater complexity. A comprehensive analysis of its impact is also presented in Section 4.3.

### 3.2.3 PROBABILISTIC TASK LOCATOR (PTL)

After continual learning, most MoE-based methods revert to a single shared router that attempts to serve all tasks. Since the router is fine-tuned on the most recent task, its decision boundary drifts toward the latest data, leading to progressive misrouting and forgetting of earlier tasks, as illustrated in Table 5 (row "Last" of the left part). Keeping one dedicated router per task (enabled by PGKE) would solve this, but we must know *which* router to activate at test time when task labels are absent. Training an additional locator (classifier) to access task IDs either (i) requires storing replay data or (ii) depends on fragile hand-crafted rules. Besides, when new tasks arrive, the classifier must be retrained or expanded, which both suffers from forgetting and exhibits poor scalability.

Drawn inspiration from (Pu et al., 2016; Pinheiro Cinelli et al., 2021; An & Cho, 2015), we propose the Probabilistic Task Locator (PTL), a *replay-free* mechanism that automatically selects the correct task's router at inference. PTL treats the hidden representation of each multimodal input as a sample from a task-specific distribution, fits each task distribution with a lightweight VAE, and scores test samples by their likelihood under each task model. This design yields a scalable task locator that requires no retraining as new tasks arrive, thereby substantially mitigating forgetting.

**Task-wise VAE fitting.** For the $i$-th task, we pass the input through the frozen LLaVA and fetch the last-token feature $\mathbf{F}_{\text{end}}$. The VAE encoder projects $\mathbf{F}_{\text{end}}$ to a Gaussian in the latent space as follows:

$$\boldsymbol{\mu}_{\text{latent}} = \text{FFN}_{\mu}^{\text{down}}(\mathbf{F}_{\text{end}}), \quad \boldsymbol{\sigma}_{\text{latent}} = \text{Softplus}\big(\text{FFN}_{\sigma}^{\text{down}}(\mathbf{F}_{\text{end}})\big), \tag{3}$$

where the Softplus ensures $\boldsymbol{\sigma}_{\text{latent}} > 0$. We draw $N_{\text{rep}}$ latent codes $\boldsymbol{z}_i \sim \mathcal{N}(\boldsymbol{\mu}_{\text{latent}}, \boldsymbol{\sigma}_{\text{latent}}^2)$ and decode them with an up-projection FFN, obtaining the predictive parameters $\big(\boldsymbol{\mu}_{\text{rec},i}, \boldsymbol{\sigma}_{\text{rec},i}\big)$ analogous to Eq. 3. The conditional likelihood of the feature can therefore be formulated as:

$$p(\mathbf{F}_{\text{end}} \,|\, \boldsymbol{z}_i) = \mathcal{N}\big(\mathbf{F}_{\text{end}}; \boldsymbol{\mu}_{\text{rec},i}, \boldsymbol{\sigma}_{\text{rec},i}^2\big) = \frac{1}{\boldsymbol{\sigma}_{\text{rec},i}\sqrt{2\pi}} \exp\!\Big[-\frac{\|\mathbf{F}_{\text{end}} - \boldsymbol{\mu}_{\text{rec},i}\|^2}{2\boldsymbol{\sigma}_{\text{rec},i}^2}\Big]. \tag{4}$$

Averaging Eq. 4 over $N_{\text{rep}}$ samples estimates the reconstruction probability $p_{\text{rec}}(\mathbf{F}_{\text{end}})$ for task $i$.

**Primitive bank construction.** After the VAE convergence, we collect the most recent $T$ training instances of task $i$, compute their $p_{\text{rec}}(\mathbf{F}_{\text{end}})$, and record the empirical mean and standard deviation, representing the primitive distribution of task $i$. Consequently, we construct a key-value bank $B = \{(\text{primitive}_i, \text{router}_i) \,|\, i = 1, \ldots, N\}$, where each key stores the probability statistics and each value is the frozen router produced by PGKE.

**Task-agnostic inference.** For an input, we obtain its feature $\mathbf{F}_{\text{end}}$, evaluate and z-score normalise its reconstruction probability under every primitive in $B$, yielding scores $\{\hat{p}^{(i)}\}_{i=1}^{N}$. The task is inferred as $i^{\star} = \arg\max_i \hat{p}^{(i)}$, and the corresponding $\text{router}_{(i^{\star})}$ is activated for subsequent processing.

### 3.3 TRAINING OBJECTIVE

For PTL, the training loss consists of two components: the reconstruction loss and the Kullback-Leibler (KL) divergence between the posterior distribution and the prior distribution of the latent space. The reconstruction loss can be measured by the negative reconstruction probability:

$$\mathcal{L}_{\text{rec}} = -p_{\text{rec}}(\mathbf{F}_{\text{end}}). \tag{5}$$

We follow the common setting of VAE that the prior distribution of the latent space is $\mathcal{N}(\mathbf{0}, \mathbf{1})$(Kingma & Welling, 2013). The posterior distribution of the latent space is $\mathcal{N}(\boldsymbol{\mu}_{\text{latent}}, \boldsymbol{\sigma}_{\text{latent}}^2)$. The KL divergence $\mathcal{L}_{\text{KL}}$ is calculated between the two distributions. For PGKE, similar to most large language models, we employ the next token prediction training schema and utilize a cross-entropy loss $\mathcal{L}_{\text{CE}}$. Besides, referring to (Fedus et al., 2022), we also add a weight balance loss $\mathcal{L}_{aux}$ for MoE training. Finally, the complete loss is composed of a weighted sum of these components:

$$\mathcal{L}_{\text{total}} = \mathcal{L}_{\text{CE}} + \lambda\mathcal{L}_{\text{KL}} + \eta\mathcal{L}_{\text{rec}} + \kappa\mathcal{L}_{\text{aux}}, \tag{6}$$

where $\lambda$, $\eta$ and $\kappa$ are the weighting coefficients for loss balance.

## 4 EXPERIMENTS AND DISCUSSIONS

### 4.1 SETUPS AND IMPLEMENTATION DETAILS.

**Datasets.** We conducted experiments on the datasets included in the CoIN (Chen et al., 2024) benchmark, which encompasses a series of eight VQA tasks. These tasks include RefCOCO (Ref) (Kazemzadeh et al., 2014), ImageNet (Deng et al., 2009), TextVQA (TQA) (Singh et al., 2019), VizWiz (Gurari et al., 2018), ScieneQA (SQA) (Saikh et al., 2022), among others. Each task varies in terms of the number of data samples, stylistic features, and domain characteristics. The training set comprises a total of 569k samples, while the testing set contains 261k samples.

**Metrics.** We adopt the metric introduced in CoIN (Chen et al., 2024), which measures the discrepancy between the model's output and the ground truth. For assessing the model's overall forgetting performance across all tasks, we utilized Backward Transfer (BWT), which evaluates the model's performance on all previous tasks after it is trained on the last task, specifically quantifies the extent of forgetting, offering insights into the model's ability to retain knowledge from previous tasks.

**Baseline Models and Methods.** Following CoIN (Chen et al., 2024), we also adopted several other representative methods based on architecture and regularization, including EWC (Schwarz et al.,

Table 1: A comprehensive comparison with baseline models and other continual learning approaches built upon LLaVA is detailed in the subsequent section. *Immediate* means performance after immediate task training. *Last* means performance after the last task training. *Mean* means the average accuracy on all eight tasks.

| Setting | Method | Accuracy on Each Task | | | | | | | | Mean↑ | BWT↑ |
|---|---|---|---|---|---|---|---|---|---|---|---|
| | | SQA | TQA | ImageNet | GQA | VizWiz | Ref | VQAv2 | OCR-VQA | | |
| Multitask | MoELoRA | 75.01 | 58.90 | 96.44 | 58.15 | 56.73 | 27.54 | 64.04 | 47.81 | 60.58 | – |
| Immediate | LoRA (Hu et al., 2022) | 75.01 | 58.36 | 96.08 | 53.87 | 56.54 | 18.32 | 63.23 | 53.96 | 59.42 | – |
| | MoELoRA (Luo et al., 2024) | 78.97 | 61.01 | 97.01 | 56.24 | 56.30 | 20.63 | **66.10** | 60.57 | 62.10 | – |
| | EWC (Schwarz et al., 2018) | **79.23** | 61.26 | 96.91 | 56.43 | 60.04 | 19.21 | 66.00 | 60.44 | 62.44 | – |
| | LWF (Li & Hoiem, 2017) | 78.83 | 61.57 | **97.07** | 56.75 | 53.48 | 20.57 | 65.27 | 61.10 | 61.83 | – |
| | MoExtend (Zhong et al., 2024) | 77.43 | 59.31 | 96.77 | **56.47** | **57.91** | 24.18 | 64.32 | 60.11 | 62.06 | – |
| | Ours | 79.01 | **59.94** | 96.85 | 56.43 | 57.44 | **25.63** | 65.15 | **62.01** | **62.81** | – |
| Last | LoRA (Hu et al., 2022) | 68.50 | 42.01 | 34.65 | 40.39 | 40.87 | 3.60 | 55.29 | 53.96 | 42.41 | -17.01 |
| | MoELoRA (Luo et al., 2024) | 67.06 | 48.16 | 36.22 | 41.83 | 41.00 | 2.62 | 56.48 | 60.57 | 44.24 | -17.86 |
| | EWC (Schwarz et al., 2018) | 60.25 | 47.92 | 30.36 | 41.33 | 31.12 | 4.53 | 56.31 | 60.44 | 41.53 | -20.90 |
| | LWF (Li & Hoiem, 2017) | 65.15 | 49.46 | 32.52 | 41.05 | 37.88 | 4.81 | 56.20 | 61.10 | 43.51 | -18.31 |
| | O-LoRA (Wang et al., 2023a) | 75.40 | 52.89 | 71.85 | 47.30 | 37.35 | 7.10 | 61.85 | 61.20 | 51.87 | -17.43 |
| | LoTA (Panda et al., 2024) | 67.30 | 41.51 | 8.25 | 37.15 | 42.25 | 0.10 | 47.95 | 56.15 | 37.58 | -17.14 |
| | SEFE (Chen et al., 2025) | 75.35 | **58.66** | 83.10 | **54.25** | 48.85 | 16.75 | **65.35** | **66.25** | 58.57 | -10.45 |
| | Ours | **77.55** | 58.17 | **94.50** | 48.91 | **55.45** | **23.40** | 56.40 | 59.44 | **59.23** | **-3.58** |

2018), LWF (Li & Hoiem, 2017), MoExtend (Zhong et al., 2024), O-LoRA (Wang et al., 2023a), LoTA (Panda et al., 2024), SEFE (Chen et al., 2025) as baselines to compare with our proposed method. It is important to note that all of these compared methods operate under the paradigm of training with clear task boundaries. Notably, EWC (Schwarz et al., 2018) also requires task IDs during the inference phase. We also reproduced the parameter expansion mechanism of Moextend (Zhong et al., 2024) as a "Immediate" Setting baseline. Given that it is not primarily focused on continual learning, we did not evaluate its "Last" setting. Furthermore, we align factors that could potentially affect fairness, such as the rank of LoRA and the data used for initializing the model. For more details, refer to CoIN (Chen et al., 2024).

**Training Details.** Our network is built upon the pretrained LLaVA. It is important to emphasize that throughout all our training processes, only the newly added experts and the corresponding routers are trained, while other existing components remain frozen. New experts' weights are copied from the experts selected by Task Probe, and new router weights are initialized by copying old routers and concatenating them with random linear weights. While training task probe, we randomly select 10% data from each task and utilize them to generate probability of each layer to extend. Please refer to the Appendix for details of the network structures, hyperparameters, and implementation details.

## 4.2 Quantitative Results on Continual Learning Benchmark

As presented in Table 1, we conducted an evaluation on the CoIN (Chen et al., 2024) Benchmark. Compared with other methods, our model demonstrates outstanding performance in both immediate and post-last-task training phases, while the average number of trainable parameters in our model is only 43.96M, more details can be found in the appendix. This significantly reduces the training cost. In the setting of *Last*, our method significantly outperforms previous approaches, demonstrating its strong capability in mitigating forgetting, especially on the ImageNet dataset, our method achieves an improvement of nearly 85%. Additionally, it is observed that on the OCR-VQA dataset, all other methods yield consistent results across both settings. However, when a new task is introduced, these methods exhibit substantial forgetting on this dataset, as evidenced by the performance in the first seven tasks. In contrast, our method maintains a stable performance of approximately 59%, showcasing its robustness against forgetting. Besides, as evident from the left of Table 2, employing our proposed method for expert extension and training yields superior results on most tasks compared to the outcomes of training without expanding experts and unfreezing the previously frozen experts.

Table 2: **Left**: Comparison of forgetting between the Extend and w/o Extend models. "Extend" freezes original experts and adds new ones using our method, while "w/o Extend" unfreezes all experts and continues training without adding new ones. **Right**: Comparison of expert addition strategies. "Ours" adds experts at selected layers using our method, "Random" adds experts randomly at the same number of layers, and "Every-layer" adds experts to all layers. Param-ratio is the proportion of parameters added by Ours compared to Every-layer.

| Dataset | w/o Extend | Extend |
|---|---|---|
| SQA | 76.68 | **77.55** |
| TQA | 49.42 | **58.17** |
| ImageNet | 45.72 | **94.50** |
| GQA | 44.65 | **48.91** |
| VizWiz | 46.79 | **55.45** |
| Ref | 4.00 | **23.40** |
| VQAv2 | **58.58** | 56.40 |
| OCR-VQA | **60.24** | 59.44 |
| BWT↑ | -17.68 | **-3.58** |

| Dataset | Random | Every-layer | Ours | Param-ratio |
|---|---|---|---|---|
| SQA | 76.73 | **80.03** | 79.01 | 0.65 |
| TQA | 58.87 | **60.07** | 59.94 | 0.812 |
| ImageNet | 96.75 | **97.09** | 96.85 | 0.75 |
| GQA | **57.60** | 57.28 | 56.43 | 0.625 |
| VizWiz | 54.62 | 56.52 | **57.44** | 0.5 |
| Ref | 20.98 | **31.68** | 25.63 | 0.75 |
| VQAv2 | 64.20 | 64.97 | **65.15** | 0.75 |
| OCR-VQA | 56.87 | 59.78 | **62.01** | 0.84 |

## 4.3 ABLATION STUDY AND ANALYSIS

***Where should the experts be extended?*** First, as shown in the left of Table 2, expanding experts is vital for learning new knowledge and confronting forgetting. When expanding experts, a common practice is to add experts to every layer for each task (Yang et al., 2024). However, we argue that this approach is inefficient due to significant knowledge overlap between tasks, which leads to parameter redundancy. To validate the effectiveness of our PGKE algorithm, as shown in Table 3, we compare several expansion strategies. By comparing the results of Ours and Every-layer, it is evident that our method achieves comparable performance with only 60% of the parameters, even outperforming the every-layer approach on OCR-VQA. Furthermore, the comparison between Random and Ours demonstrates that the positions identified by the task probe are more reasonable, yielding an average performance improvement of 3%-4% under the same training parameter budget, showing the superiority of the PGKE method.

***How does task order influence performance?*** We conducted experiments on our method under different training orders across eight datasets of CoIN to verify its robustness to training order. As shown in the table, the forgetting level of our method remains stable across different training orders. This is because the feature extraction and reconstruction of different tasks are independent, same as task routers. Previously trained tasks do not affect the features of subsequent tasks. Additionally, training order has a minimal impact on the method's immediate performance, though this is not our primary focus. More details will be discussed in the Appendix.

Table 3: Comparison of different task orders. "Norm" denotes that we follow CoIN's training order. "Rev" denotes that we reverse CoIN's training order. "Rand" denotes that we random shuffle the training order of the CoIN's datasets.

| Setting | Immediate | | | Last | | |
|---|---|---|---|---|---|---|
| | Norm | Rev | Rand | Norm | Rev | Rand |
| ScienceQA | 79.01 | **79.96** | 79.63 | 77.55 | 78.42 | **78.48** |
| TextVQA | **59.94** | 58.60 | 58.51 | **58.17** | 56.12 | 56.01 |
| ImageNet | 96.85 | **96.99** | 96.89 | 94.50 | **94.61** | 94.55 |
| GQA | 56.43 | **57.88** | 57.81 | 48.91 | **50.32** | 50.29 |
| VizWiz | 57.44 | **58.76** | 57.92 | 55.45 | **56.77** | 56.70 |
| Ref | 25.63 | **29.29** | 29.13 | 23.40 | **27.23** | 27.09 |
| VQAv2 | **65.15** | 64.75 | 64.83 | **56.40** | 56.02 | 56.16 |
| OCR-VQA | **62.01** | 61.27 | 61.31 | **59.44** | 58.72 | 58.66 |
| Mean ↑ | 62.81 | **63.43** | 63.25 | 59.23 | **59.78** | 59.74 |
| BWT ↑ | - | - | - | -3.58 | -3.66 | **-3.51** |

***How many experts should be extended?*** Our experiments show that for certain challenging tasks, increasing the number of parameters is crucial, even when added to layers with overlapping knowledge (see the left side of Table 2). To investigate further, we examined how the number of experts per layer affects performance on the Ref and SQA tasks by adding 1, 4, or 8 experts to probe-selected layers (results in Table 4). We found that performance on the more difficult Grounding task improves significantly with more parameters, while the simpler SQA task benefits only marginally. However,

due to the distinction between task difficulty and task dissimilarity, we could not develop an end-to-end method for determining the optimal number of experts based on task difficulty. As a result, we focus on optimizing layer selection and treat the number of experts as a hyperparameter.

***How does the PTL mechanism perform?*** To figure out the PTL mechanism, we conducted the following two ablation experiments: 1) **Last** task evaluation: We utilized the router and the corresponding set of experts learned from the last task to evaluate all previous tasks. 2) **Random** task evaluation: We randomly selected a task router and its corresponding set of experts to evaluate all tasks. Results are presented in the left part of Table 5. The last task is excluded since it is not affected by forgetting.

Table 4: The impact of the number of experts added at specified layers on the final performance.

| Number of Added Experts | 1 | 4 | 8 |
|---|---|---|---|
| Ref | 25.63 | 35.13 | 41.51 |
| ScienceQA | 79.01 | 81.42 | 82.01 |

First, using the either the last or the random task's router results in catastrophic forgetting w.r.t the routers of previous tasks. In contrast, our PTL learns a easily-scalable task locator that adaptively select tasks' specific routers, better retain knowledge of past tasks. Additional experiments on feature extraction methods are detailed in the Appendix.

***Can knowledge from previous tasks facilitate new task learning?*** In continual learning, beyond mitigating forgetting, it is also important to assess whether prior knowledge facilitates learning new tasks. To this end, we evaluated the model's forward transfer by comparing it to models trained from scratch on each of the eight tasks, with an equal number of trainable parameters, as shown in the right side of Table 5. Results show that while prior knowledge offers limited benefits for simpler tasks, it significantly accelerates learning on more challenging tasks like Ref and OCRVQA.

Figure 4: Qualitative Results of LLaVA-CMoE.

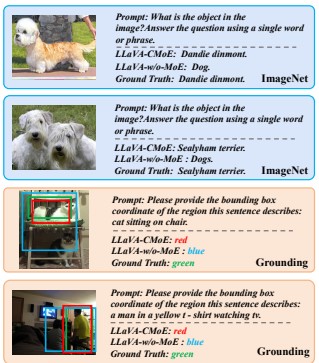

Table 5: Left: Comparison of different task classification strategies. Right: Knowledge forward transfer ability comparison.

| Dataset | Last | Random | Ours | Dataset | Separate | Ours |
|---|---|---|---|---|---|---|
| SQA | 73.03 | 61.92 | **77.55** | SQA | 78.97 | **79.01** |
| TQA | 46.82 | 50.62 | **58.17** | TQA | **60.56** | 59.94 |
| ImageNet | 29.68 | 36.59 | **94.50** | ImageNet | **97.05** | 96.85 |
| GQA | 41.81 | 43.12 | **48.91** | GQA | 56.31 | **56.43** |
| VizWiz | 44.32 | 37.86 | **55.45** | VizWiz | 56.20 | **57.44** |
| Ref | 9.92 | 4.83 | **23.40** | Ref | 21.3 | **25.63** |
| VQAv2 | 55.13 | 45.90 | **56.40** | VQAv2 | 65.01 | **65.15** |
| OCR | 62.01 | 25.62 | **59.44** | OCR | 60.79 | **62.01** |

***Initialization of New Expert.*** To determine the optimal strategy, we conducted a comprehensive ablation study comparing four distinct initialization methods: (1) Zero Initialization, (2) Average Initialization, (3) Initialization from Probe Experts, and (4) Initialization from the Nearest Expert (our proposed method, inheriting weights from the most semantically similar expert of the previous task).

As shown in Table 6, initializing from the nearest expert yields the superior performance on the majority of datasets (e.g., TextVQA, ImageNet, Ref, OCRVQA). We attribute this success to a "warm start" mechanism: in the early stages of training, a new expert initialized with valid prior knowledge (from a similar old expert) has a significantly higher probability of being selected by the router compared to a randomly or averagely initialized one. This ensures the expert receives sufficient

Table 6: Performance Comparison of Different Initialization Strategies

| Initialization Strategy | SQA | TQA | ImageNet | GQA | VizWiz | Ref | VQAv2 | OCR | Mean |
|---|---|---|---|---|---|---|---|---|---|
| **Zero Init** | 77.42 | 57.13 | 93.22 | 55.39 | 55.32 | 23.12 | 64.13 | 60.39 | 60.77 |
| **Average Init** | 78.33 | 57.94 | 95.37 | 55.25 | **57.72** | 24.62 | 64.30 | **62.16** | 61.96 |
| **Probe Expert Init** | **79.13** | 59.44 | 96.51 | **56.55** | 57.29 | 25.27 | **65.21** | 61.77 | 62.65 |
| **Ours (Nearest Expert)** | 79.01 | **59.94** | **96.85** | 56.43 | 57.44 | **25.63** | 65.15 | 62.01 | **62.81** |

Table 7: Performance and Expert Addition Comparison with/without Load Balancing

| Setting | Metric | SQA | TQA | ImageNet | GQA | VizWiz | Ref | VQAv2 | OCR | Avg. Added |
|---|---|---|---|---|---|---|---|---|---|---|
| w/o Load Balancing | Accuracy (%) | **79.05** | 59.89 | 96.65 | 56.32 | **57.52** | **25.75** | **65.18** | **62.22** | - |
| | Experts Added | 24 | 27 | 24 | 21 | 18 | 27 | 26 | 27 | 24.25 |
| Ours (with LB) | Accuracy (%) | 79.01 | **59.94** | **96.85** | **56.43** | 57.44 | 25.63 | 65.15 | 62.01 | - |
| | Experts Added | **21** | **26** | **24** | **20** | **16** | **24** | **24** | **27** | **22.75** |

gradient updates early on, allowing it to rapidly adapt and eventually differentiate its capabilities to fit the new task requirements. We have incorporated these comparative results into the revised manuscript to provide a rigorous justification for our design choice.

***Will probe activation inflated by load-balancing loss?*** Theoretically, since probe experts are initialized from an average distribution without strong priors, they are at a disadvantage compared to the well-trained, frozen experts. Without the LB loss, the router is susceptible to unstable convergence, often resulting in a "winner-take-all" scenario or inefficient routing decisions.

To empirically verify this, we conducted an ablation study removing the Load Balancing (LB) loss during the probe stage. As shown in the Table 7, removing the LB loss actually results in a higher rate of expert expansion (e.g., adding 24 experts vs. 21 on SQA, and 18 vs. 16 on VizWiz) without yielding significant performance gains. This indicates that without the regularization provided by the LB loss, the router tends to blindly allocate new parameters. Conversely, the inclusion of LB loss effectively "rationalizes" the selection process; it prevents unnecessary expansion by enforcing a fairer probability distribution, which, in practice, encourages the model to leverage the capabilities of existing experts rather than defaulting to new ones. Thus, the LB loss serves as a critical regularizer to minimize redundant parameter growth while maintaining optimal performance.

**Qualitative Results.** We qualitatively analyze the model's outputs. As illustrated in Figure 4, after training on the final task, we randomly sample data from previous tasks and compare results across methods. On ImageNet, our model retains domain-specific knowledge with minimal forgetting, while LLaVA-w/o-MoE relies on pretrained knowledge and produces generic responses. For the Grounding task, our model better preserves knowledge required for non-linguistic generation.

## 5  CONCLUSION AND DISCUSSION

In this paper, we propose LLaVA-CMoE, a continual learning framework comprising two key modules: Probe-Guided Knowledge Extension (PGKE) and Probabilistic Task Locator (PTL). PGKE addresses the inefficiency of continuous parameter expansion by adaptively increasing parameters through probe-guided expert addition. Meanwhile, PTL mitigates catastrophic forgetting in continual learning by modeling task distributions and memorizing the mapping between task distributions and router networks. Qualitative and quantitative results demonstrate that our method remarkably outperforms the existing methods.

**Limitation**. Expert addition is currently limited, as adding experts only to the language model may not fully benefit tasks requiring detailed visual understanding. Additionally, increasing tasks raises storage demands, and existing distillation or merging methods can cause router-related forgetting. We will address these issues in future work.

**Societal impact**. Our method improves continual learning in multimodal models, benefiting applications like education and accessibility. However, increased adaptability may also heighten risks of misuse, underscoring the need for responsible deployment and oversight.

# 6 ETHICS STATEMENT

**Data Usage** Experiments relied on the public CoIN benchmark (8 VQA tasks like RefCOCO, ImageNet). No personally identifiable, sensitive, or proprietary data was used; human subjects were not involved, meeting data compliance requirements.

**Bias Mitigation** No new annotations or external knowledge were added, preserving result objectivity by avoiding extraneous biases.

# 7 REPRODUCIBILITY STATEMENT

To ensure the reproducibility of this research, the detailed method is described in Section 3, hyperparameters for all models are detailed in Appendix A.3, and all training data, as well as code will be publicly released within the conference scope.

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

## A  TECHNICAL APPENDICES AND SUPPLEMENTARY MATERIAL

**Algorithm of Probe-Guided Knowledge Extension (PGKE).** The pseudo-code of our algorithm is shown in Algorithm 1.

---
**Algorithm 1:** Probing and Training Task $\mathcal{T}_i$

---
**Input:** Original model $\pi$, probing training data $D_{\mathrm{p}}^{\mathrm{train}}$, probing eval data $D_{\mathrm{p}}^{\mathrm{eval}}$, full training data $D^{\mathrm{train}}$, threshold coefficient $\alpha$

$\pi_0 \leftarrow \mathrm{clone}(\pi)$
$probe\_layers \leftarrow \mathrm{range}(N_{layer})$
$\mathrm{append\_expert}(\pi_0, probe\_layers)$
$\pi_0 \leftarrow \mathrm{train}(\pi_0, D_{\mathrm{p}}^{\mathrm{train}})$
$prob\_freq \leftarrow \mathrm{compute\_per\_layer\_freq}(\pi_0, D_{\mathrm{p}}^{\mathrm{test}})$
**for** $layer \leftarrow 0$ **to** $prob\_layers - 1$ **do**
  $\mu \leftarrow \mathrm{mean}(prob\_freq[layer])$
  $\sigma \leftarrow \mathrm{std}(prob\_freq[layer])$
  $freq \leftarrow prob\_freq[layer][-1]$
  **if** $freq > \mu - \alpha \cdot \sigma$ **then**
    $selected\_layer.\mathrm{append}(layer)$
**end**
$\mathrm{append\_expert}(\pi, selected\_layer)$
$\pi \leftarrow \mathrm{train}(\pi, D^{\mathrm{train}})$

---

### A.1  MORE RESULTS

**Visualization of PTL mechanism.** We evaluated the PTL mechanism on test data across eight tasks, and the results are shown in Figure 5. As observed from the test results, the PTL mechanism achieves localization accuracies exceeding 80% on five tasks: ScienceQA, ImageNet, VizWiz, Grounding and OCR-VQA. Besides, we can also find that the localization performance on the remaining three tasks, GQA, TextVQA, and VQAv2, is relatively weaker compared to the other five tasks. Through our analysis, this is attributed to the significant overlap in the features of images and questions across these tasks. For instance, GQA and VQAv2 share high similarities in terms of question formats, image content, and styles, which leads to a substantial portion of VQAv2 samples being "mislocalized" to the GQA task.

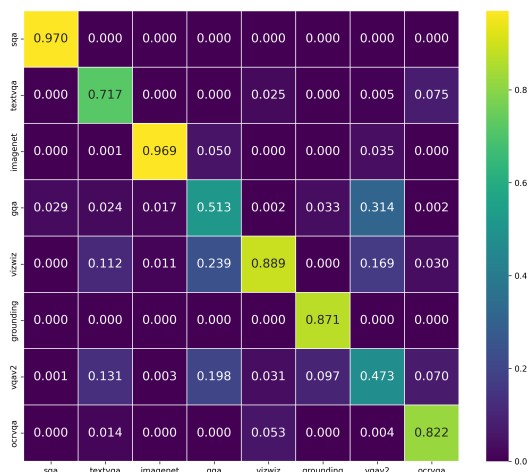

Figure 5: **Confusion Matrix of PTL.** This figure illustrates the localization performance of the PTL mechanism, where the data in each row and column represent the frequency with which the test data corresponding to the task represented by the column is assigned to the task represented by the row.

**Impact of selection threshold $\alpha$ in PGKE.** We conducted an ablation experiment on the threshold $\alpha$ in PGKE, and the results are shown in Table 8. As observed from the results, with a strict threshold (e.g., $\alpha = 0.4$), while the average parameter growth is reduced, the network exhibits a slight performance decline in the early training stages. Particularly, in the later training stages, after the network has accumulated substantial knowledge, a strict threshold increases the difficulty for the network to capture the distributional differences between the current task and previously accumulated knowledge. This causes the network to tend to avoid adding new parameters, resulting in significant performance degradation on tasks such as OCR-VQA and VQAv2. Similarly, when the threshold is set to a loose value (e.g., $\alpha = 1.2$), the model's performance approaches that of the every-layer setting, and no substantial parameter

Table 8: Comparison of different selection threshold $\alpha$ on model's performance.

| $\alpha$ or setting | Accuracy on Each Task | | | | | | | | Avg-Acc | Avg-Params |
| --- | --- | --- | --- | --- | --- | --- | --- | --- | --- | --- |
| | SQA | TQA | ImageNet | GQA | VizWiz | Ref | VQAv2 | OCR-VQA | | |
| 0.8 | 79.01 | 59.94 | 96.85 | 56.43 | 57.44 | 25.63 | 65.15 | 62.01 | 62.81 | 0.71 |
| 0.4 | 78.85 | 59.63 | 96.81 | 55.59 | 56.95 | 24.64 | 62.05 | 59.54 | 61.72 | 0.59 |
| 1.2 | 79.57 | 59.95 | 96.90 | 56.52 | 57.42 | 28.31 | 65.22 | 61.34 | 63.15 | 0.84 |
| every-layer | 80.03 | 60.07 | 97.09 | 57.28 | 56.52 | 31.68 | 64.97 | 59.78 | 63.43 | 1.00 |

Table 9: Comparison of different feature selection's influence on model's performance.

| Method | Accuracy on Each Task | | | | | | | | Mean↑ | BWT↑ |
| --- | --- | --- | --- | --- | --- | --- | --- | --- | --- | --- |
| | SQA | TQA | ImageNet | GQA | VizWiz | Ref | VQAv2 | OCR-VQA | | |
| Last Layer's feature | 77.55 | 58.17 | 94.50 | 48.91 | 55.45 | 23.40 | 56.40 | 59.44 | 59.23 | -3.58 |
| Pooled feature | 77.67 | 59.19 | 89.03 | 51.06 | 57.03 | 24.69 | 63.43 | 20.68 | 55.34 | -15.31 |

Table 10: Comparison of our method on llava-v1.5-7B and llava-v1.5-13B.

| Setting | Model Size | Accuracy on Each Task | | | | | | | | Mean↑ | BWT↑ |
| --- | --- | --- | --- | --- | --- | --- | --- | --- | --- | --- | --- |
| | | SQA | TQA | ImageNet | GQA | VizWiz | Ref | VQAv2 | OCR-VQA | | |
| Immediate | 7B | 79.01 | 59.94 | 96.85 | 56.43 | 57.44 | 25.63 | 65.15 | 62.01 | 62.81 | – |
| | 13B | 82.32 | 65.77 | 98.26 | 60.41 | 62.35 | 30.91 | 68.75 | 67.92 | 67.08 | – |
| Last | 7B | 77.55 | 58.17 | 94.50 | 48.91 | 55.45 | 23.40 | 56.40 | 59.44 | 59.23 | -3.58 |
| | 13B | 80.79 | 64.82 | 96.01 | 52.93 | 60.86 | 29.11 | 62.18 | 62.27 | 63.62 | -3.46 |

savings are achieved. Thus, after evaluating the excessively low performance associated with $\alpha = 0.4$ and the excessive parameter count of $\alpha = 1.2$, we selected $\alpha = 0.8$ as a more balanced value.

**Impact of different selection of PTL feature.** In our main results, we chose the output of the last Decoder layer for classification in the PTL. In Table 9, we also attempted to use the average pooling of outputs from all Decoder layers for classification. Classifying using features from average pooling slightly improved performance on several tasks (SQA, TQA, GQA, VizWiz, Ref), particularly achieving a 7.03% improvement on VQAv2. However, severe performance degradation ($\sim$39%) occurred when facing OCR-VQA. We further explored the causes of this decline and found that most samples were classified into the ImageNet task, while a small number of ImageNet samples were classified into OCR-VQA, indicating that average pooling is poor in distinguishing highly similar tasks.

**Impact of model size.** We conducted experiments on models of varying scales, as delineated in Table 10. Owing to the augmentation in the parameter count of the base model, the performance of the proposed method was further enhanced. Notably, the BWT metric of our approach exhibits a higher value on the 13B model. This phenomenon may be attributed to the fact that the output of the final decoder layer in larger models encompasses more highly abstracted information (hidden-size: 5120 in the 13B model vs. 4096 in the 7B model), which facilitates PTL in achieving superior task classification performance.

**Impact of LoRA-rank.** We further investigated the impact of the LoRA rank on our method, as presented in the Table 11. When the LoRA rank was set to 32, there remained a certain margin for performance improvement in the model. However, when the LoRA rank reached 128, the model performance had reached a plateau, with minor performance degradation observed on specific tasks. Therefore, we selected a LoRA rank of 64 as the default setting in our method.

**Comparison with replay methods.** To ensure a more rigorous comparison, we extended the baseline method, where a specialist is added to all layers for each new task, with a replay-based scheme. Specifically, when learning the Nth task, we sample 10% of the data from all previous tasks (1 to N-1) and mix it with the current task's data for training. The experimental results are shown in Table 12. It can be observed that after incorporating additional data for training, the replay-based scheme does

Table 11: Comparison of different lora rank.

| Setting | Rank | Accuracy on Each Task | | | | | | | | Mean↑ | BWT↑ |
|---------|------|------|------|----------|------|--------|------|-------|--------|-------|------|
| | | SQA | TQA | ImageNet | GQA | VizWiz | Ref | VQAv2 | OCR-VQA | | |
| Immediate | 32 | 78.86 | 59.90 | 96.32 | 55.92 | 57.06 | 24.17 | 63.15 | 60.22 | 61.95 | – |
| | 64 | 79.01 | 59.94 | 96.85 | 56.43 | 57.44 | 25.63 | 65.15 | 62.01 | 62.81 | – |
| | 128 | 79.23 | 59.54 | 97.07 | 57.96 | 56.03 | 25.58 | 65.22 | 62.33 | 62.75 | – |
| Last | 32 | 77.32 | 58.11 | 94.06 | 48.44 | 55.07 | 22.08 | 54.02 | 57.80 | 58.36 | -3.59 |
| | 64 | 77.55 | 58.17 | 94.50 | 48.91 | 55.45 | 23.40 | 56.40 | 59.44 | 59.23 | -3.58 |
| | 128 | 77.70 | 57.76 | 94.59 | 50.06 | 54.12 | 23.11 | 56.49 | 59.66 | 59.19 | -3.56 |

Table 12: Comparison with replay methods.

| Setting | replayed | Accuracy on Each Task | | | | | | | | Mean↑ | BWT↑ | Training-time |
|---------|----------|------|------|----------|------|--------|------|-------|--------|-------|------|---------------|
| | | SQA | TQA | ImageNet | GQA | VizWiz | Ref | VQAv2 | OCR-VQA | | | |
| Immediate | Yes | 79.84 | 60.17 | 97.02 | 57.27 | 56.59 | 32.11 | 64.88 | 61.92 | 63.73 | – | 20h |
| | No (ours) | 79.01 | 59.94 | 96.85 | 56.43 | 57.44 | 25.63 | 65.15 | 62.01 | 62.81 | – | 17h |
| Last | Yes | 77.93 | 58.11 | 94.16 | 50.32 | 53.52 | 28.04 | 60.81 | 60.48 | 62.81 | -3.36 | 20h |
| | No (ours) | 77.55 | 58.17 | 94.50 | 48.91 | 55.45 | 23.40 | 56.40 | 59.44 | 59.23 | -3.58 | 17h |

Table 13: Comparison of the average trainable parameters (M) over eight datasets, and performance between LoRA, MoELoRA, EWC, LWF and Ours.

| Metrics | LoRA | MoELoRA | EWC | LWF | Ours |
|---------|------|---------|-----|-----|------|
| #Trainable Parameters per Task | 31M | 62M | 62M | 62M | 43.96M |
| Mean↑ | 42.41 | 44.24 | 41.53 | 43.51 | 59.23 |
| BWT↑ | -17.01 | -17.86 | -20.90 | -18.31 | -3.58 |

further improve model performance. However, it also indeed increases both data storage overhead and computational overhead.

**Parameter comparsion.** We tabulated the number of parameters for different methods, as shown in Table 13. It is worth noting that the LoRA method we compared only adds LoRA matrices to the up-proj layer. Although the LoRA method has the fewest trainable parameters, it exhibits poor anti-forgetting performance. Our method adaptively adds experts based on task differences, significantly enhancing anti-forgetting capabilities while maintaining minimal parameter growth. The changes in trainable parameters across different tasks are illustrated in Figure 6.

**Qualitative Results.** We qualitatively analyze the model's outputs. As illustrated in Figure 7, after training on the final task, we randomly sample data from previous tasks and compare results across methods. On ImageNet, our model retains domain-specific knowledge with minimal forgetting, while MoELoRA and other methods rely on pretrained knowledge and produces generic responses. For the Ref task, our model better preserves knowledge required for non-linguistic generation.

**Performance over tasks.** We evaluate our methods' performance during the entire training process. As shown in Table 14, our approach maintains excellent anti-forgetting capability after training on sequential tasks, particularly on the SQA, TQA, ImageNet, VizWiz, and Ref datasets. By comparison, PTL exhibits slightly weaker performance on the VQAv2 and GQA datasets due to a slight classification confusion issue. Nevertheless, its performance remains comparable to or better than existing methods, as demonstrated in our main manuscript.

**Expansion of experts per layer over tasks.** For each task, we recorded the layers expanded during each extension in Table 15. Since the number of expanded layers exceeds that of non-expanded layers, for convenience, we list the non-expanded layers for each task: It can be observed that the model adds significantly fewer experts to the first 16 layers (low-level features) and predominantly expands parameters in the last 16 layers (high-level features). This aligns with our intuition, as the model shares some low-level features and mainly adds high-level knowledge.

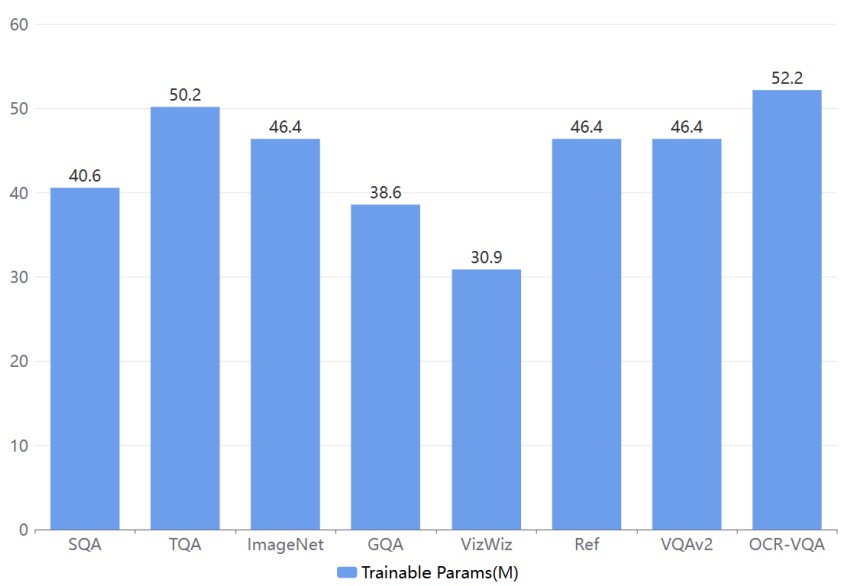

Figure 6: The added trainable parameters on eight tasks.

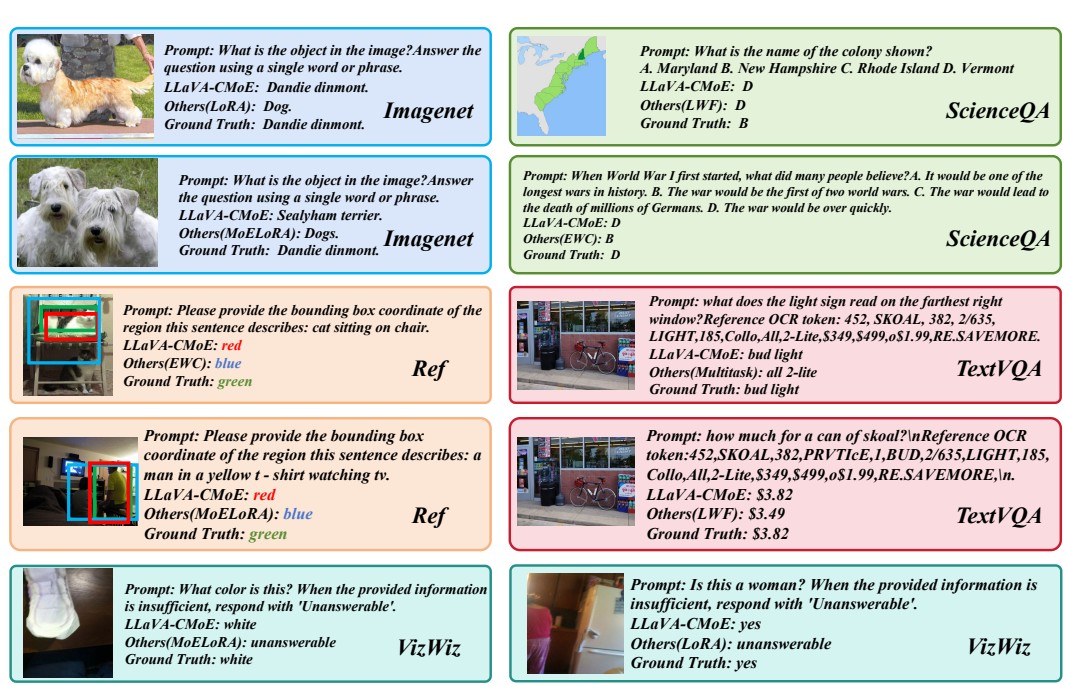

Figure 7: We randomly select five datasets, which are relatively prone to forgetting, and visualized the results of the two models for comparison.

## A.2 CLARIFICATION

**Clarification of settings.** Our experiments adopt the CoIN benchmark, which is constructed under the task-incremental learning (Task-IL) setting. Our approach, however, differs in that it achieves task-agnostic inference and data replay-free training on this foundation. Furthermore, CoIN is a benchmark with explicit task boundaries, and we also take this attribute as a default assumption during training. This constitutes a prior that is widely relied upon by state-of-the-art methods in the field. If no task boundary partitioning is implemented during training, severe inter-task interference will be induced. Consequently, all methods (including baseline models) will suffer a drastic performance drop, and the corresponding results are listed on Table 1 in the main text.

Table 14: The performance on eight datasets during sequential training. Training order: SQA → TQA → ImageNet → GQA → VizWiz → Ref → VQAv2 → OCR-VQA.

| Training Dataset | Accuracy on Each Task | | | | | | | |
|---|---|---|---|---|---|---|---|---|
| | SQA | TQA | ImageNet | GQA | VizWiz | Ref | VQAv2 | OCR-VQA |
| SQA | 79.01 | - | - | - | - | - | - | - |
| TQA | 79.01 | 59.94 | - | - | - | - | - | - |
| ImageNet | 79.01 | 59.71 | 96.85 | - | - | - | - | - |
| GQA | 79.00 | 59.28 | 96.31 | 56.43 | - | - | - | - |
| VizWiz | 78.21 | 58.59 | 95.13 | 55.86 | 57.44 | - | - | - |
| Ref | 78.21 | 58.58 | 95.12 | 54.40 | 57.42 | 25.63 | - | - |
| VQAv2 | 77.73 | 58.31 | 94.51 | 49.86 | 56.31 | 24.02 | 65.15 | - |
| OCR-VQA | 77.55 | 58.17 | 94.50 | 48.91 | 55.45 | 23.40 | 56.40 | 62.01 |

Table 15: The scaling of per-layer experts as tasks expand.

| Tasks | Layer ids not expand expert |
|---|---|
| SQA | 1, 2, 3, 4, 6, 9, 10, 14, 15, 23, 24 |
| TQA | 0, 2, 3, 4, 5, 29 |
| ImageNet | 0, 1, 3, 6, 7, 8, 19, 21 |
| GQA | 0, 2, 3, 6, 7, 10, 11, 13, 15, 24, 25, 27 |
| VizWiz | 2, 4, 5, 6, 8, 10, 11, 14, 15, 16, 19, 20, 25, 28, 30, 31 |
| Ref | 1, 4, 8, 12, 14, 17, 23, 28 |
| VQAv2 | 3, 4, 5, 12, 16, 23, 27, 31 |
| OCR-VQA | 1, 7, 1, 14, 27 |

## A.3 TRAINING DETAILS

**Experiments environment.**

We train the model using $8 \times$ H20 GPUs. It takes 2 hours to train the initial eight experts, followed by 15 hours for the continual learning process. Throughout the training, we set the warmup ratio to 0.03, use the AdamW optimizer, and employ torch BF16 precision with DeepSpeed stage zero2. The rank of LoRA experts is set to 64, the rank alpha to 128, and the global batch size to 128. We assign a weight of 1e-3 to the load balancing loss of the router, the KL divergence loss, and the reconstruction loss. For training the normal experts, we use a learning rate of 2e-4, and during the probe experts training, we increase the learning rate to 3e-4.

Table 16: Time consumption of each process.

| Tasks | Probe training(min) | Probe test(min) | Task training(min) |
|---|---|---|---|
| SQA | 4 | 1.5 | 25 |
| TQA | 7 | 2.6 | 41 |
| ImageNet | 15 | 2.1 | 113 |
| GQA | 23 | 2.8 | 174 |
| VizWiz | 6 | 1.8 | 30 |
| Ref | 21 | 3 | 169 |
| VQAv2 | 22 | 3.1 | 175 |
| OCR-VQA | 25 | 1.8 | 182 |

**Duration of each processes.** We present statistics on the training and inference time of the task probe in Table 16. It can be observed that, on 8 H20 GPUs, each task probe process only accounts for 15% of the model training time, which falls within an acceptable time range.

## A.4 THE USE OF LARGE LANGUAGE MODELS.

In this work, we utilized large language models (LLMs) exclusively for grammatical refinement and minor linguistic touch-ups. Every edit supported by LLMs underwent meticulous review and

verification by the authors, ensuring that no fabricated content or unintended distortions of the original meaning were introduced. Notably, the research concepts, experimental design, data analysis, and conclusions presented in this study were entirely conceived and implemented by the authors, with no assistance from LLMs.

