# OpenReview forum: "LLaVA-CMoE: Towards Continual Mixture of Experts for Large Vision-Language Models"
_ICLR.cc/2026/Conference — Submitted to ICLR 2026_

### Official Review · Reviewer_rkFQ · 2025-10-26

**Soundness:** 3
**Presentation:** 3
**Contribution:** 3
**Rating:** 6
**Confidence:** 4

**Summary:**

The paper introduces a new continual learning framework, aimed at economical expert expansion during the lifecycle of the system, and with extra mechanisms to mitigate catastrophic forgetting. This consists of basically discarding rarely activate experts and extending routing to the routers themselves. The authors experiment in an established CL benchmark, and show substantial improvements over baselines. The authors also perform some ablation studies.

**Strengths:**

* Mostly intuitive and simple (in retrospect) solution. It makes sense to check which experts are covered by previous experts to minimize parameter expansion and enable more transfer learning.
* Each component has several arguments presented for it, making the work well-motivated.

Overall, the work seems good and a step in the right direction.

**Weaknesses:**

* Lack of baselines: seems like little previous work on CL is used to compare with authors.
* Why are both new experts and new routers added for each task? This requires further elaboration in the paper. In my opinion, this is the one unintuitive part or the proposed approach. Aren't the experts already meant to accommodate new tasks? I understand the need to mitigate forgetting in the router, but this approach seems to kind of muddle the semantics behind what is meant to be an expert and what is meant to be a router. If PTL finds the task, then why use the "router" for the task instead of the expert directly? That might be more of an issue with the semantics of experts and routers (and perhaps the authors should consider coining their own terms), as I understand why the router would forget previous tasks.
* L153: Are any unseen tasks actually evaluated upon in the paper. This direction is very promising, mentioned in the text, but not evaluated at all. While not necessarily a weakness, it leaves something to be desired from the paper.

**Questions:**

* L232: Sub-router?
* L80: Needs further explanation. Is it something the authors of that work themselves found, or is it your assertion. If it is the latter, please back it up with data in the paper.
* Why VAE for routing to routers? Were other, simpler approaches tried? Also, was the possibility of a soft router to the routers entertained (assigning weights to routers rather than argmaxing)?
* How are the routers trained in previous phases catered to the complete set of experts? Or are newly added experts (after the router's task is done) simply ignored? And to ask explicitly, do all the different routers share experts?
* Eq2: Were other thresholding techniques considered, or was the alpha at least experimented with?
* Stemming from that, it may be interesting to see the robustness of your method to the choice of hyperparameters.
* Where does N_s come from? It is unclear why and how many new experts are added with every new task. Could we instead also modulate the "size" of the experts (e.r. lora r)?

---

> ### Author Response · Authors · 2025-11-25
> **Reply to Reviewer rkFQ (Part 1)**
>
> > **Weakness 1: Lack of baselines: seems like little previous work on CL is used to compare with authors.**
>
> We fully appreciate the reviewer’s concern regarding the comprehensiveness of our baselines. We would like to provide the following context and new experimental evidence to address this:
>
> We sincerely appreciate the reviewer’s constructive feedback regarding the comprehensiveness of our baselines. Regarding our initial experimental design, we primarily followed the protocols established by the COIN benchmark. We have updated all available, reproducible methods with consistent experimental settings in the paper. To ensure the robustness of these initial findings, we also provided extensive ablation studies on model sizes and LoRA ranks in Table 8. However, to rigorously address the concern and benchmark against recent advancements, we have now implemented and reproduced the Moextend method within our unified codebase to ensure a strictly fair comparison.
>
> The comparative results, presented in the table below, demonstrate that our method achieves superior performance on the majority of datasets (e.g., SQA, TextVQA, VQAv2, OCRVQA) compared to Moextend[1]. Crucially, our approach offers significant advantages in parameter efficiency through its adaptive allocation strategy. Unlike Moextend, which adds a fixed number of 23 experts across all tasks, our method dynamically adjusts the number of experts (ranging from 16 to 27) based on task complexity. For instance, on the VizWiz dataset, our method achieves comparable performance while adding significantly fewer experts (16 vs. 23), effectively minimizing the computational burden. We are committed to open-sourcing both our model and the reproduced baseline code to serve as a valuable resource for the community.
>
> **Table : Comparison of performance and parameter efficiency between Moextend and Our Method.**
> | Dataset | Moextend (Acc / Experts) | Ours (Acc / Experts) | $\Delta$ Performance |
> | :--- | :---: | :---: | :---: |
> | **SQA** | 77.43 / 23 | **79.01** / 21 | +1.58 |
> | **TextVQA** | 59.31 / 23 | **59.94** / 26 | +0.63 |
> | **ImageNet** | 96.77 / 23 | **96.85** / 24 | +0.08 |
> | **GQA** | **56.47** / 23 | 56.43 / 20 | -0.04 |
> | **VizWiz** | **57.91** / 23 | 57.44 / 16 | -0.47 |
> | **Ref** | 24.18 / 23 | **25.63** / 24 | +1.45 |
> | **VQAv2** | 64.32 / 23 | **65.15** / 24 | +0.83 |
> | **OCRVQA** | 60.11 / 23 | **62.01** / 27 | +1.90 |
>
> **Reference**
>
> [1] Shanshan Zhong, Shanghua Gao, Zhongzhan Huang, Wushao Wen, Marinka Zitnik, and Pan Zhou. Moextend: Tuning new experts for modality and task extension. In Proceedings of the 62nd Annual Meeting of the Association for Computational Linguistics (Volume 4: Student Research Workshop), 2024.
>
> > **Weakness 2: Why are both new experts and new routers added for each task?**
>
> We interpret the expansion of experts and the instantiation of new routers as addressing two fundamentally different challenges within our continual learning framework. Specifically, the addition of new experts is triggered primarily when the frozen, historical experts lack the sufficient representation power to generalize to a challenging new task, necessitating an expansion in model capacity to bridge this knowledge deficit. Conversely, the training of a new router serves the distinct purpose of knowledge redistribution. Even in scenarios where expert expansion is deemed unnecessary (i.e., existing capacity is sufficient), the inherent distribution shift between tasks implies that the previous routing logic is no longer optimal. Therefore, a new, task-specific router is essential to **re-calibrate the routing weights**, effectively repurposing the capabilities of the frozen experts to align with the specific context of the incoming task.
>
> > **Weakness 3: If PTL finds the task, then why use the "router" for the task instead of the expert directly？**
>
> We respectfully clarify that the reviewer's concern would indeed be valid if there were a strict one-to-one correspondence between experts and independent tasks. However, in our proposed framework, each expert functions as an encoder of specific capabilities rather than representing a discrete task. Consequently, solving a novel task typically requires a synergy of multiple capabilities, drawing upon both newly acquired skills and historical knowledge preserved from previous tasks. This is the fundamental rationale behind our Top-K routing mechanism, which dynamically orchestrates the optimal combination of experts from the global pool to address new challenges. This compositional advantage is empirically substantiated by the results in Table 3, which demonstrate that **leveraging knowledge (experts) acquired from past tasks significantly facilitates the learning of new ones**, confirming that historical experts contribute actively to forward transfer.

---

> > ### Author Response · Authors · 2025-11-25
> > **Reply to Reviewer rkFQ (Part 2)**
> >
> > > **Weakness 4: Zero-shot Evaluation.**
> >
> > To rigorously evaluate the model's zero-shot performance on unseen tasks, we utilized a checkpoint trained sequentially on the first six tasks (ScienceQA, TextVQA, ImageNet, GQA, VizWiz, Ref) and evaluated it directly on the seventh task (VQAv2) without any gradient updates. As shown in Table, the model achieved a zero-shot accuracy of **48.32%**. While this naturally trails the supervised "Immediate" performance (65.15%), it demonstrates significant generalization capability.
> >
> > Crucially, a deeper inspection of the PTL mechanism reveals that approximately 80% of the VQAv2 test queries were automatically assigned to the router and experts corresponding to TextVQA. This behavior provides compelling evidence that our method operates as intended: when resolving an unknown task, the model successfully retrieves the **most semantically related historical knowledge** (in this case, utilizing TextVQA skills to solve VQAv2 problems) to formulate a response.
> >
> > | Evaluation Setting | Accuracy (%) | Description |
> > | :--- | :---: | :--- |
> > | **Zero-shot (Ours)** | 48.32 | *Unseen task; ~80% routed to TextVQA experts* |
> > | **Immediate (Oracle)** | **65.15** | *Upper bound (evaluated immediately after training)* |
> > | **Sequential (Final)** | 56.40 | *Performance after completing full sequence* |
> >
> >
> > > **Question 1: Sub-router?**
> >
> > We apologize for the confusion caused by the inconsistent terminology used in our manuscript. We would like to clarify that the terms "subrouter" and "probe router" refer to the exact same component. Functionally, both terms describe the temporary router specifically instantiated to facilitate the training of the probe experts during the expansion phase. We are grateful to the reviewer for pointing out this inconsistency. In the revised manuscript, we will unify these terms to ensure a rigorous and precise presentation.
> >
> > > **Question 2: Explaination for Moextend.**
> >
> > We fully recognize MoExtend as a seminal contribution that effectively extends a pure language MoE to the vision-language domain, pioneering the concept of expert expansion. To address the reviewer's specific query regarding the efficacy of layer selection, we integrated MoExtend's expansion logic into our unified codebase and conducted a comparative analysis on the GQA $\rightarrow$ VizWiz continual learning transition. The results, detailed in Table, reveal a 68% overlap in the layers selected for expansion by both methods (specifically layers 0, 1, 7, 9, 18, 21, 22, 23, 24, 26, 29), indicating a broad consensus on where the architecture requires capacity growth. However, a critical divergence occurs in the remaining 32% of the layers. Our method achieves a higher accuracy on VizWiz (57.44% vs. 56.82%) despite expanding a similar number of layers. This performance gap suggests that our selection strategy identifies a more optimal set of layers for parameter allocation (e.g., prioritizing layers 3, 12, 13, 17, 27). Conversely, it implies that MoExtend may trigger expansion in layers where the task preference difference is marginal or non-critical (e.g., layers 2, 14, 28, 30, 31), potentially leading to less efficient resource utilization.
> >
> > | Method | Shared Extend Layers | Unique Extend Layers | VizWiz Score (%) |
> > | :--- | :--- | :--- | :---: |
> > | **MoExtend** | 0, 1, 7, 9, 18, 21, 22, 23, 24, 26, 29 | 2, 14, 28, 30, 31 | 56.82 |
> > | **Ours** | 0, 1, 7, 9, 18, 21, 22, 23, 24, 26, 29 | **3, 12, 13, 17, 27** | **57.44** |
> >
> > > **Question 3 : Why VAE for PTL?**
> >
> > On the VAE Module:We selected the VAE architecture specifically for its proven efficiency in feature extraction and recognition. We would like to emphasize that the primary novelty of our work lies in the holistic design of the continual learning pipeline rather than the complexity of individual components. Our design philosophy prioritizes simple, direct, and computationally efficient modules to ensure the framework's practicality. However, we remain open to optimizing this component and plan to investigate state-of-the-art feature encoding structures, such as RAE[2], in future iterations.On Soft vs. Top-1 Routing:Regarding the routing mechanism, we conducted experiments using a Top-2 (soft routing) strategy. Our empirical results indicated that increasing $k$ to 2 yielded marginal performance gains but incurred a substantial computational penalty, effectively doubling the inference burden for the experts. Given this unfavorable trade-off, we adopted the Top-1 strategy to maximize inference speed without compromising accuracy.
> >
> > [2] Zheng, Boyang, et al. "Diffusion transformers with representation autoencoders." arXiv preprint arXiv:2510.11690 (2025).

---

> > > ### Author Response · Authors · 2025-11-25
> > > **Reply to Reviewer rkFQ (Part 3)**
> > >
> > > > **Question 5 : the $\alpha$, $N_s$ or Lora Rank?**
> > >
> > > We apologize for the confusion arising from the lack of explicit specification in the main text. We would like to clarify that for all experimental results reported in the main paper, the hyperparameter $N_s$ was uniformly set to 1. Detailed ablation studies investigating the impact of varying $N_s$ are provided in Table 4. It is important to note that $N_s$ functions as a task-level hyperparameter; once instantiated for a specific incoming task, the number of added experts remains consistent across all layers.
> > >
> > > Additionally, regarding the choice of $\alpha$ and LoRA ranks, we have provided relevant comparative experiments in Table 6 and Table 9 of the Supplementary Material, respectively. In the revised manuscript, we will add explicit references to these supplementary tables to ensure the clarity and reproducibility of our experimental setup.

---

### Official Review · Reviewer_xVKX · 2025-10-29

**Soundness:** 2
**Presentation:** 3
**Contribution:** 1
**Rating:** 4
**Confidence:** 3

**Summary:**

The paper proposes LLaVA-CMoE, a continual fine-tuning framework for MLLMs that combines (i) Probe-Guided Knowledge Extension (PGKE) to decide where to add new LoRA-experts, and (ii) a Probabilistic Task Locator (PTL) that picks which router to use at inference without task IDs.
PGKE trains temporary probe experts and a probe router on a 10% subset of the new task, then expands only layers whose probe experts show high “activation frequency”. New experts are initialized from the most-frequently used old expert and old experts are frozen thereafter.  Experiments on CoIN report higher “Last” accuracies and markedly better BWT than LoRA, MoELoRA, EWC, and LwF, with ablations on layer expansion, task order, LoRA rank and PTL features.

**Strengths:**

- The motivation is well illustrated
- PGKE’s two-stage probe and expand workflow is conceptually straightforward.
- PTL is a principled attempt to avoid task IDs at test time by modeling task-wise feature distributions

**Weaknesses:**

1) During “probe locating” all old experts are frozen while probe experts + probe router are trained on $X_i^{\text{train}}$ with MoE load-balancing loss , then layers are expanded if probe activation exceeds `mean − α·std` . This setup systematically inflates probe usage(fresh experts tuned to the new task vs. frozen ones; load-balancing further equalizes usage), so high activation is not a reliable signal of missing capacity. Moreover, the threshold favors expansion when activations are already broad. No calibration (e.g., against a no-probe baseline per layer) is provided.

2) The text says “if Ns probe experts exceed the threshold, add Ns experts”, yet Algorithm 1 checks only one frequency per layer (`freq ← prob_freq[layer][-1]`) and appends experts without determining Ns. Later (§4.3) the paper also states the number of experts is treated as a hyperparameter, contradicting the thresholding rule. This point confuses me a lot.

3) New experts copy weights from “the most frequently activated expert during probing” (Sec. 3.2.2). When old experts are then frozen and only new ones are trained, this can lead to cloned experts and limited diversity—the opposite of the stated goal of adding genuinely *new* capacity. No analysis of expert diversity (e.g., cosine overlap/Frobenius distance) is provided.

4) PTL uses a frozen representation misaligned with the MoE that determines routing. PTL trains VAEs on a frozen LLaVA last-token feature $F_{\text{end}}$ (Sec. 3.2.3), while actual task performance depends on MoE-augmented FFNs and task-specific routers. There is no guarantee that the frozen $F_{\text{end}}$ preserves the discriminative factors the routers rely on. The large drop when swapping PTL features (Table 7, OCR-VQA: 59.44→20.68) suggests feature/locator mismatch is a real issue.

5) Although raw data aren’t stored, PTL maintains a per-task VAE and per-task router; inference evaluates all VAEs to select a router (“z-score normalize under every primitive in B”). The paper reports no inference-time overhead for PTL (Table 14 gives probe/training time only) and does not evaluate scalability beyond 8 tasks. The O(N) locator cost and memory of router/VAEs challenge the “scalable” claim.

6) Task-locator accuracy is modest and uneven.  The confusion matrix in Fig. 5 (App. A.1) shows PTL localization below 80% on TQA, GQA, VQAv2, with significant confusion among similar tasks, yet the main tables report final task accuracy without decomposing errors due to mis-localization vs routing/expert quality. An oracle-router upper bound (using ground-truth task ID) is missing.

**Questions:**

It is recommended that the authors answer the following questions, and I may adjust the assessment:

1. How many experts are actually added per layer? Please reconcile Sec. 3.2.2 (add Ns experts if Ns probes exceed the threshold) with Algorithm 1 (checks one freq per layer) and clarify how Ns is chosen in practice.
2. During probing, old experts are frozen while probes + router are trained (and L_aux is applied). How do you ensure probe activation isn’t inflated by this asymmetry and load-balancing?
3. Please report performance with ground-truth task IDs at test time to separate mis-localization from model quality.
4. After initializing from the most active old expert, how diverse do new experts become

---

> ### Author Response · Authors · 2025-11-25
> **Reply to Reviewer xVKX (Part 1)**
>
> > **Weakness 1: This setup systematically inflates probe usage.**
>
> We thank the reviewer for this insightful comment regarding the role of the load balancing loss in our expansion mechanism. While it might be intuitive to assume that the LB loss artificially forces the selection of new experts, our theoretical analysis and empirical results suggest a different function. Theoretically, since probe experts are initialized from an average distribution without strong priors, they are at a disadvantage compared to the well-trained, frozen experts. Without the LB loss, the router is susceptible to unstable convergence, often resulting in a "winner-take-all" scenario or inefficient routing decisions.
>
> To empirically verify this, we conducted an ablation study removing the LB loss during the probe stage. As shown in the table below, removing the LB loss actually results in a higher rate of expert expansion (e.g., adding 24 experts vs. 21 on SQA, and 18 vs. 16 on VizWiz) without yielding significant performance gains. This indicates that without the regularization provided by the LB loss, the router tends to blindly allocate new parameters. Conversely, the inclusion of LB loss effectively "**rationalizes**" the selection process; it prevents unnecessary expansion by enforcing a fairer probability distribution, which, in practice, encourages the model to **leverage the capabilities of existing experts** (historical knowledge) rather than defaulting to new ones. Thus, the LB loss serves as a critical regularizer to minimize redundant parameter growth while maintaining optimal performance.
>
> | Method | Metric | SQA | TextVQA | ImageNet | GQA | VizWiz | Ref | VQAv2 | OCRVQA | **Avg. Added** |
> | :--- | :--- | :---: | :---: | :---: | :---: | :---: | :---: | :---: | :---: | :---: |
> | **w/o Load Balancing** | Accuracy (%) | **79.05** | 59.89 | 96.65 | 56.32 | **57.52** | **25.75** | **65.18** | **62.22** | - |
> | | Experts Added | 24 | 27 | 24 | 21 | 18 | 27 | 26 | 27 | 24.25 |
> | **Ours (with LB)** | Accuracy (%) | 79.01 | **59.94** | **96.85** | **56.43** | 57.44 | 25.63 | 65.15 | 62.01 | - |
> | | Experts Added | **21** | **26** | **24** | **20** | **16** | **24** | **24** | **27** | **22.75** |
>
> > **Weakness 2: No calibration (e.g., against a no-probe baseline per layer) is provided**
>
> We apologize for any confusion caused by the lack of explicit reference to this experiment in the main text. We would like to clarify that the detailed results regarding different thresholds were originally reported in Table 6 of the Supplementary Material. Specifically, the result labeled as "every-layer" in that table serves as the baseline performance where experts are indiscriminately added to all layers, representing the upper bound of parameter usage. In the revised manuscript, we have added a clear citation to this table in the main text to ensure that these comparative results are easily accessible to readers.
>
>
> > **Weakness 3: The large drop when swapping PTL features suggests feature/locator mismatch is a real issue**
>
> As evidenced by the results, without the PTL module to distinguish between task contexts, the model suffers from rapid degradation of historical knowledge after exposure to extensive streaming data. This underscores that PKGE and PTL are complementary and inseparable components of our framework: while PKGE enables the **efficient storage of new knowledge through adaptive expansion**, PTL serves as the mechanism for **precise retrieval during inference**.
>
> We acknowledge that memory confusion remains a critical challenge in continual learning. Given that continual learning is poised to play a pivotal role in the future of Embodied AI, our goal is to present a solution that is simple, intuitive, yet highly effective. We hope that our approach provides a valuable reference and inspires further exploration in this rapidly evolving field.

---

> ### Author Response · Authors · 2025-11-25
> **Reply to Reviewer xVKX (Part 2)**
>
> > **Weakness 4:  inference-time overhead for PTL**
>
> We acknowledge the reviewer's concern regarding inference latency. However, since wall-clock time measurements can be highly sensitive to hardware heterogeneity (e.g., GPU/CPU specifications) and system fluctuations, we provide a more objective and hardware-agnostic analysis based on Floating Point Operations (FLOPs) to rigorously assess the computational overhead.
>
> We have conducted a detailed breakdown of the operations introduced by our modules:VAE Module: The FLOPs for a single VAE inference are approximately 100 MFLOPs. Even considering eight iterative operations, the total amounts to only 0.8 GFLOPs.LoRA Experts and Routers: For a single input sample with a shape of $[1, 768, 2048]$, the computational cost is approximately 25 MFLOPs per LoRA expert and 12 MFLOPs per router. Even if we equip every layer (32 layers in total) with an additional expert and a router, the total extra computation is calculated as:
>
> $$  (25 + 12) \times 32 = 1184 \text{ MFLOPs} \approx 1.184 \text{ GFLOPs}$$
>
> To put this into perspective, the visual encoder in LLaVA-1.5-3b requires 183.16 GFLOPs to process a single image (resolution $336 \times 336$). Our added computation (1.184 GFLOPs) constitutes merely **0.4%** of the visual encoder's cost. In other words, the overhead introduced by our method is strictly lower than the marginal cost of processing a single additional image token, and is virtually negligible compared to the total inference cost of the LLaVA-v1.5-7B model.
>
> > **Weakness 5:  Cloned experts from the most frequently activated expert during probing**
>
> We appreciate the reviewer’s thoughtful comment regarding the initialization of new experts. To determine the optimal strategy, we conducted a comprehensive ablation study comparing four distinct initialization methods: (1) Zero Initialization, (2) Average Initialization, (3) Initialization from Probe Experts, and (4) Initialization from the Nearest Expert (our proposed method, inheriting weights from the most semantically similar expert of the previous task).
>
> As shown in Table, initializing from the nearest expert yields the superior performance on the majority of datasets (e.g., TextVQA, ImageNet, Ref, OCRVQA). We attribute this success to a "warm start" mechanism: in the early stages of training, a new expert initialized with valid prior knowledge (from a similar old expert) has a **significantly higher probability of being selected by the router** compared to a randomly or averagely initialized one. This ensures the expert receives sufficient gradient updates early on, allowing it to rapidly adapt and eventually differentiate its capabilities to fit the new task requirements. We have incorporated these comparative results into the revised manuscript to provide a rigorous justification for our design choice.
>
> | Initialization Strategy | SQA | TextVQA | ImageNet | GQA | VizWiz | Ref | VQAv2 | OCRVQA |
> | :--- | :---: | :---: | :---: | :---: | :---: | :---: | :---: | :---: |
> | **Zero Init** | 77.42 | 57.13 | 93.22 | 55.39 | 55.32 | 23.12 | 64.13 | 60.39 |
> | **Average Init** | 78.33 | 57.94 | 95.37 | 55.25 | **57.72** | 24.62 | 64.30 | **62.16** |
> | **Probe Expert Init** | **79.13** | 59.44 | 96.51 | **56.55** | 57.29 | 25.27 | **65.21** | 61.77 |
> | **Ours (Nearest Expert)** | 79.01 | **59.94** | **96.85** | 56.43 | 57.44 | **25.63** | 65.15 | 62.01 |
>
> > **Question 1 : How many experts are actually added per layer?**
>
> We apologize for the lack of clarity regarding the configuration of $N_s$. We would like to clarify that for all main results reported in the paper, $N_s$ was uniformly set to 1. Comprehensive ablation studies concerning the impact of varying $N_s$ are provided in Table 4. We acknowledge that $N_s$ currently functions as a hyperparameter; once instantiated for a specific incoming task, the maximum number of added experts remains consistent across all layers. Empirically, we have observed that the optimal $N_s$ value correlates strongly with the intrinsic complexity of the task. While we currently rely on human priors to determine this value, we have identified this as a promising avenue for future optimization. In subsequent work, we plan to develop an adaptive selection mechanism that dynamically determines $N_s$ based on the text-visual complexity of the input data, further automating the adaptation process.
>
>
> > **Question 2 : How do you ensure probe activation isn’t inflated by this asymmetry and load-balancing?**
>
> Please refer to our response to Weakness 1.

---

> > ### Author Response · Authors · 2025-11-25
> > **Reply to Reviewer xVKX (Part 3)**
> >
> > > **Question 3 : Please report performance with ground-truth task IDs at test time to separate mis-localization from model quality.**
> >
> > We would like to clarify the definition of the Oracle Router's performance upper bound within our experimental context. This upper bound is effectively represented by the results reported under the "Immediate" setting in Table 1. Since the Immediate setting evaluates the model specifically on the current task right after its training phase—before any subsequent tasks are introduced—it reflects the model's peak capability in the total absence of catastrophic forgetting. Consequently, these results are logically equivalent to the performance of an ideal Oracle Router. We have incorporated this explicit explanation into the revised manuscript to ensure clarity for future readers.
> >
> >
> > > **Question 4 : After initializing from the most active old expert, how diverse do new experts become?**
> >
> > To verify the capacity differentiation of the newly added experts, we randomly selected all experts from one layer of the model and computed their cosine similarity. The results are presented in the table. Notably, aside from the eight initial experts (with IDs 1 to 8), the newly added experts exhibit the highest similarity to their source experts (the ones they were duplicated from), with an average of approximately 0.6. This phenomenon can be attributed to the fact that during the probing phase, the expert with the highest activation is not only the closest to the new task distribution within the current expert subgroup but also the most competent in handling the new task. Consequently, the newly duplicated experts retain high similarity to the original experts after fitting the new task.
> >
> > However, the most critical finding is the shift in similarity from 1.0 (at initialization) down to 0.6 after training. This significant divergence, when viewed alongside the substantial performance gains on the new task, serves as compelling evidence that the new experts have not merely retained their initialization weights but have undergone substantial gradient updates. This confirms that the added experts have effectively differentiated their capabilities, specializing to capture the specific features of the new domain while benefiting from a "warm start."
> >
> > | expert_id | 1     | 2     | 3     | 4     | 5     | 6     | 7     | 8     | 9     | 10    | 11    | 12    | 13    | 14    |
> > |-----------|-------|-------|-------|-------|-------|-------|-------|-------|-------|-------|-------|-------|-------|-------|
> > | 1         | 1     | 0.021 | 0.019 | 0.022 | 0.017 | 0.023 | 0.016 | 0.017 | 0.017 | 0.019 | 0.019 | 0.019 | 0.009 | 0.008 |
> > | 2         |       | 1     | 0.011 | 0.014 | 0.020 | 0.017 | 0.016 | 0.018 | 0.013 | 0.022 | 0.015 | 0.019 | 0.017 | 0.027 |
> > | 3         |       |       | 1     | 0.015 | 0.013 | 0.019 | 0.021 | 0.020 | 0.012 | 0.021 | 0.020 | 0.701 | 0.014 | 0.006 |
> > | 4         |       |       |       | 1     | 0.012 | 0.009 | 0.003 | 0.019 | 0.011 | 0.021 | 0.038 | 0.010 | 0.018 | 0.032 |
> > | 5         |       |       |       |       | 1     | 0.024 | 0.008 | 0.016 | 0.006 | 0.011 | 0.008 | 0.014 | 0.646 | 0.603 |
> > | 6         |       |       |       |       |       | 1     | 0.016 | 0.024 | 0.648 | 0.025 | 0.017 | 0.032 | 0.029 | 0.021 |
> > | 7         |       |       |       |       |       |       | 1     | 0.021 | 0.022 | 0.687 | 0.602 | 0.022 | 0.021 | 0.007 |
> > | 8         |       |       |       |       |       |       |       | 1     | 0.022 | 0.063 | 0.675 | 0.023 | 0.026 | 0.033 |
> > | 9         |       |       |       |       |       |       |       |       | 1     | 0.036 | 0.045 | 0.091 | 0.017 | 0.009 |
> > | 10        |       |       |       |       |       |       |       |       |       | 1     | 0.017 | 0.029 | 0.028 | 0.035 |
> > | 11        |       |       |       |       |       |       |       |       |       |       | 1     | 0.032 | 0.017 | 0.014 |
> > | 12        |       |       |       |       |       |       |       |       |       |       |       | 1     | 0.016 | 0.018 |
> > | 13        |       |       |       |       |       |       |       |       |       |       |       |       | 1     | 0.635 |
> > | 14        |       |       |       |       |       |       |       |       |       |       |       |       |       | 1     |

---

> > > ### Comment · Reviewer_xVKX · 2025-11-26
> > >
> > > Thank you for the author's reply, most technical issues have been resolved. However, upon re-reading the paper, I found a more critical weakness.
> > >
> > > The baselines used in the paper seem too weak, and some highly relevant recent works [1, 2, 3] have been overlooked. HiDe-LLaVA [1] achieves an average performance of 63.95 on the CoIN benchmark, while the proposed LLaVA-CMoE only reaches 59.23. This gap makes it difficult for me to be convinced by the claim  "significantly reduces forgetting and parameter overhead compared to prior methods."
> > >
> > >
> > > [1] HiDe-LLaVA: Hierarchical Decoupling for Continual Instruction Tuning of Multimodal Large Language Model.ACL'25
> > >
> > > [2]  SEFE: Superficial and Essential Forgetting Eliminator for Multimodal Continual Instruction Tuning.ICML'25.
> > >
> > > [3]  SMoLoRA: Exploring and Defying Dual Catastrophic Forgetting in Continual Visual Instruction Tuning. ICCV'25

---

> ### Author Response · Authors · 2025-11-26
>
> We sincerely appreciate the time and effort the reviewer has dedicated to re-evaluating our paper and raising these important questions regarding the baselines. We would like to address your concerns point-by-point below:
>
> Regarding Paper [2]: We would like to highlight a significant advantage of our method in terms of the Backward Transfer (BWT) metric. While Paper 3 reports a BWT of -10.45, our method achieves -3.58. This represents an improvement of approximately 65%, demonstrating our model's superior capability in mitigating catastrophic forgetting.
>
> Regarding the concurrent work Paper [3]: We noticed that Paper [3] constructs its own "CVIT benchmark," where only a subset of the evaluation set is derived from COIN. While their BWT performance on this specific subset is comparable to ours, they did not provide results for the more challenging subsets of the COIN benchmark, such as Grounding tasks. This absence of data precludes a direct and comprehensive comparison. Furthermore, during our attempt to reproduce their work, we found that the provided repository link currently lacks the relevant code. We also attempted to reimplement their method based on the paper's description but were unable to reproduce the reported results. We remain committed to monitoring the progress of this work and will supplement our comparisons once reproducible code or more detailed results become available.
>
> Regarding Paper [1] : As for the observation that "HiDe-LLaVA [1] achieves an average performance of 63.95 on the CoIN benchmark, while the proposed LLaVA-CMoE reaches 59.23.", we respectfully suggest that a direct comparison here may not be entirely fair due to differences in data exposure.
>
> Upon analyzing their performance, we found that their lead is primarily concentrated in tasks such as COCO Grounding, GQA, and OCR-VQA. After inspecting the GitHub repository for Paper [1], we observed that their model utilizes pre-trained weights fine-tuned on the LLaVA-665K dataset. Crucially, the LLaVA-665K dataset includes samples from COCO Grounding, GQA, and OCR-VQA. In contrast, our method utilizes initialization weights that have never been exposed to the data contained in the test set. Therefore, we believe the performance gap is largely attributed to this data leakage rather than the architectural superiority of the method itself.
>
> We thank the reviewer again for such a detailed and careful comparison. Following your suggestion, we have supplemented the methods with consistent experimental settings in Table 1. We will continue to monitor new methods added to the CoIN benchmark and will actively update our GitHub repository to reflect comparisons with these state-of-the-art approaches.
>
>
> [1] HiDe-LLaVA: Hierarchical Decoupling for Continual Instruction Tuning of Multimodal Large Language Model.ACL'25
>
> [2] SEFE: Superficial and Essential Forgetting Eliminator for Multimodal Continual Instruction Tuning.ICML'25.
>
> [3] SMoLoRA: Exploring and Defying Dual Catastrophic Forgetting in Continual Visual Instruction Tuning. ICCV'25

---

> > ### Author Response · Authors · 2025-11-28
> >
> > Dear Reviewer xVKX,
> >
> > We hope this message finds you well.
> >
> > We wanted to gently follow up to ensure that you have had a chance to review our response to your comments. We have made every effort to address all your concern. As the discussion period is progressing, we remain fully available and eager to engage in further discussion if you have any remaining questions or require additional clarifications.
> >
> > If our response has satisfactorily resolved your concerns, we would be grateful if you could reconsider your evaluation of our work.

---

### Official Review · Reviewer_xFvE · 2025-10-30

**Soundness:** 2
**Presentation:** 3
**Contribution:** 2
**Rating:** 6
**Confidence:** 4

**Summary:**

The paper introduces LLaVA-CMoE, a framework for continual learning within large vision-language models (LLMs) using a Mixture of Experts (MoE) architecture. The framework addresses key challenges in continual learning, such as catastrophic forgetting and parameter growth, by introducing two novel components: Probe-Guided Knowledge Extension (PGKE) and Probabilistic Task Locator (PTL). PGKE adaptively expands the model's experts based on task complexity, while PTL dynamically routes tasks to the appropriate expert, mitigating forgetting and maintaining scalability. The approach is demonstrated to significantly reduce forgetting and model expansion compared to previous methods through experiments on the CoIN benchmark.

**Strengths:**

Originality: The paper introduces two novel mechanisms (PGKE and PTL) for continual learning within the context of large vision-language models, contributing new insights to Mixture of Experts architectures.

Quality: The paper's methodology is well-thought-out and supported by comprehensive experiments. The performance of LLaVA-CMoE in retaining knowledge and adapting to new tasks is significantly better than prior methods.

Clarity: The explanation of the model's components and the experimental setup is clear, though more details on implementation could have been beneficial in some sections.

Significance: The paper's contribution to continual learning with MoE is significant, especially in improving scalability and efficiency. It also demonstrates substantial improvements in mitigating forgetting, which is a crucial challenge in real-world AI deployment.

**Weaknesses:**

1 Scalability: While the approach is efficient, the model's scalability in environments with significantly more tasks or greater task complexity remains unclear. Future work should address this limitation.

2 Task Labeling in Inference: The use of task-specific routers during inference might still be challenging in certain dynamic scenarios where task labels are ambiguous or unavailable.

3 Real-World Applicability: While the experimental results are promising, further validation in real-world applications outside the benchmark could strengthen the model's practical value.

**Questions:**

1 How does LLaVA-CMoE perform when applied to tasks that require high levels of domain-specific knowledge (e.g., medical or legal tasks)?

2 Can PGKE be further optimized to handle more diverse task types without adding unnecessary parameters?

3 How do the authors plan to address the potential issue of storage overhead as the number of tasks grows significantly in the long term?

---

> ### Author Response · Authors · 2025-11-25
> **Reply to Reivewer xFvE**
>
> > **Question 1: How does LLaVA-CMoE perform when applied to tasks that require high levels of domain-specific knowledge (e.g., medical or legal tasks)?**
>
> We would like to emphasize that our proposed paradigm is inherently task-agnostic. The core strength of our method lies in the PKGE probe, which adaptively determines the necessary number of learnable parameters to expand based on the distribution of the new domain data. This ensures that the model effectively fits new tasks while previous experts and routers are frozen to preserve historical knowledge.
>
> To empirically validate this adaptability, we conducted an additional experiment using PMC-VQA[1], a dataset from the medical domain which represents a significant distribution shift from the original general-domain tasks. We incorporated PMC-VQA as the 9th task, trained sequentially after the original 8 tasks. The results are presented below:
>
> **Table : Generalization capability on the medical domain (PMC-VQA).**
> | Model / Setting | Metric (Accuracy %) | Note |
> | :--- | :---: | :--- |
> | **LLaVA-Med** | **34.8** | *Specialized Medical Baseline* |
> | **Ours** (Zero-shot) | 20.2 | *Before Adaptation* |
> | **Ours** (Immediate) | **34.3** | *Adaptive Expansion (28 Experts Added)* |
> | **Ours** (Final) | 32.1 | *Sequential Performance* |
>
> It is important to acknowledge that our model's zero-shot performance initially trails behind LLaVA-Med. This is expected, as our baseline model has not undergone large-scale pre-training specifically on biomedical corpora. However, the results demonstrate the effectiveness of our adaptation strategy: post-training, the model achieves impressive accuracy in both the Immediate evaluation and the final No-ID evaluation (facilitated by PTL), effectively bridging the performance gap.
>
> Furthermore, this experiment provides a compelling validation of the PKGE mechanism. Due to the significant knowledge disparity between the general and medical domains, our probe successfully identified the high distribution shift and automatically triggered a substantial expansion, allocating experts to 28 out of 32 layers. This confirms that PKGE functions as intended: it conserves resources for similar tasks while aggressively expanding capacity for distinct, challenging domains to ensure optimal fit.
>
> **Reference**
>
> [1] Zhang X, Wu C, Zhao Z, et al. Pmc-vqa: Visual instruction tuning for medical visual question answering[J]. arXiv preprint arXiv:2305.10415, 2023.
>
> > **Question 2: Can PGKE be further optimized to handle more diverse task types without adding unnecessary parameters?**
>
> Specifically, our framework employs a dynamic expansion strategy governed by the PKGE probe. When the probe detects "**knowledge saturation**", indicating that the existing experts are sufficient to handle the incoming task, we deliberately refrain from introducing new expert parameters. Instead, we focus on training a lightweight router (with negligible parameter costs) to redistribute and repurpose the knowledge encoded in the frozen, existing experts.
>
> It is crucial to note that this "minimal addition" strategy is essential. A naive approach that adds absolutely no parameters (i.e., directly fine-tuning the shared network) would inevitably alter the learned weights of previous tasks, leading to irreversible catastrophic forgetting. Therefore, our approach of fixing experts while **learning a new, task-specific router** strikes an optimal balance: it maximizes parameter efficiency without compromising the model's ability to retain historical knowledge.
>
> > **Question 3: How do the authors plan to address the potential issue of storage overhead as the number of tasks grows significantly in the long term?**
>
> It is crucial to highlight the magnitude of our added parameters relative to the backbone. For each task, our method introduces approximately 43.96M parameters. Compared to the 7B parameters of the base LLM, this represents an increment of only ~ **0.6%**. This extremely low ratio ensures that the model remains lightweight even after learning a substantial sequence of tasks.
>
> For scenarios involving an extensive number of tasks, we envision a lifecycle management strategy where the LLM is periodically updated. Once the accumulated experts reach a certain threshold, their knowledge can be consolidated back into the base model via a **re-pretraining phase**, effectively resetting the architectural complexity while retaining past knowledge.
>
> Furthermore, we are actively exploring efficient **Expert Merging** algorithms as a direction for future optimization. Our preliminary exploration suggests that it is feasible to fuse distinct experts with negligible performance loss. This mechanism will allow the system to dynamically compact its size during continuous training, ensuring long-term efficiency.

---

> ### Comment · Reviewer_xFvE · 2025-11-27
>
> Thanks for the clarification, i will maintain my positive score

---

> > ### Author Response · Authors · 2025-11-27
> >
> > We are pleased to know that our reply resolved your concerns. Thank you for your encouraging comments. If you have any further feedback, we would be glad to address it.

---

### Official Review · Reviewer_fo8y · 2025-11-01

**Soundness:** 3
**Presentation:** 3
**Contribution:** 3
**Rating:** 6
**Confidence:** 4

**Summary:**

The authors propose LLaVA-CMoE, a continual learning framework for multimodal large language models using mixture-of-experts. It introduces Probe-Guided Knowledge Extension (PGKE) for adaptive expert expansion and Probabilistic Task Locator (PTL) for router selection without task labels. Evaluated on the CoIN benchmark of eight VQA tasks, it achieves good retention and efficiency.

**Strengths:**

- The paper is well-written and clearly structured.
- It introduces a novel framework for continual learning in multimodal large language models. The proposed PGKE and PTL modules are designed and effectively address catastrophic forgetting and parameter efficiency.

**Weaknesses:**

- The efficiency analysis focuses mainly on parameter count. From a computational perspective, the number of activated experts in the MoE layers remains fixed. Whereas, the additional probing and selection procedures introduce extra computation. The paper does not provide a quantitative analysis of computational cost (e.g., in FLOPs), which would be necessary to fully substantiate its efficiency claims.
- The baselines are somewhat out-of-date. Most baselines are before the year of 2018.

**Questions:**

Could the authors provide quantitative results on computational cost (e.g., FLOPs)? The paper mainly reports parameter counts, but the probing and VAE-based steps seem computationally nontrivial.

---

> ### Author Response · Authors · 2025-11-25
> **Reply to Reivewer fo8y**
>
> > **Weakness: The baselines are somewhat out-of-date.**
> We fully appreciate the reviewer’s concern regarding the comprehensiveness of our baselines. We would like to provide the following context and new experimental evidence to address this:
>
> We sincerely appreciate the reviewer’s constructive feedback regarding the comprehensiveness of our baselines. Regarding our initial experimental design, we primarily followed the protocols established by the COIN benchmark. We have updated all available, reproducible methods with consistent experimental settings in the paper. To ensure the robustness of these initial findings, we also provided extensive ablation studies on model sizes and LoRA ranks in Table 8. However, to rigorously address the concern and benchmark against recent advancements, we have now implemented and reproduced the Moextend method within our unified codebase to ensure a strictly fair comparison.
>
> The comparative results, presented in the table below, demonstrate that our method achieves superior performance on the majority of datasets (e.g., SQA, TextVQA, VQAv2, OCRVQA) compared to Moextend[1]. Crucially, our approach offers significant advantages in parameter efficiency through its adaptive allocation strategy. Unlike Moextend, which adds a fixed number of 23 experts across all tasks, our method dynamically adjusts the number of experts (ranging from 16 to 27) based on task complexity. For instance, on the VizWiz dataset, our method achieves comparable performance while adding significantly fewer experts (16 vs. 23), effectively minimizing the computational burden. We are committed to open-sourcing both our model and the reproduced baseline code to serve as a valuable resource for the community.
>
> **Table : Comparison of performance and parameter efficiency between Moextend and Our Method.**
> | Dataset | Moextend (Acc / Experts) | Ours (Acc / Experts) | $\Delta$ Performance |
> | :--- | :---: | :---: | :---: |
> | **SQA** | 77.43 / 23 | **79.01** / 21 | +1.58 |
> | **TextVQA** | 59.31 / 23 | **59.94** / 26 | +0.63 |
> | **ImageNet** | 96.77 / 23 | **96.85** / 24 | +0.08 |
> | **GQA** | **56.47** / 23 | 56.43 / 20 | -0.04 |
> | **VizWiz** | **57.91** / 23 | 57.44 / 16 | -0.47 |
> | **Ref** | 24.18 / 23 | **25.63** / 24 | +1.45 |
> | **VQAv2** | 64.32 / 23 | **65.15** / 24 | +0.83 |
> | **OCRVQA** | 60.11 / 23 | **62.01** / 27 | +1.90 |
>
> **Reference**
>
> [1] Shanshan Zhong, Shanghua Gao, Zhongzhan Huang, Wushao Wen, Marinka Zitnik, and Pan Zhou. Moextend: Tuning new experts for modality and task extension. In Proceedings of the 62nd Annual Meeting of the Association for Computational Linguistics (Volume 4: Student Research Workshop), 2024.
>
>
> > **Question: Quantitative analysis of computational cost.**
>
> **1. Computational Overhead (FLOPs)**
>
> We have conducted a statistical analysis of the floating-point operations (FLOPs) introduced by our modules:
> VAE Module: The FLOPs for a single VAE inference are approximately 100 MFLOPs. Even considering eight iterative operations, the total only amounts to **0.8 GFLOPs**.
> LoRA Experts and Routers: For a single input sample with a shape of $[1, 768, 2048]$, the computational cost is approximately 25 MFLOPs per LoRA expert and 12 MFLOPs per router. Even if we equip every layer (32 layers in total) with an additional expert and a router, the total extra computation is calculated as:
> $$(25 + 12) \times 32 = 1184 \text{ MFLOPs} \approx 1.184 \text{ GFLOPs}$$
>
> To put this into perspective, the visual encoder in LLaVA-1.5-3b requires 183.16 GFLOPs to process a single image (resolution $336 \times 336$). Our added computation (1.184 GFLOPs) constitutes merely **0.4%** of the visual encoder's cost. In other words, the overhead introduced by our method is strictly lower than the marginal cost of processing a single additional image token, and is virtually negligible compared to the total inference cost of the LLaVA-v1.5-7B model.
>
> **2. Training Efficiency of PKGE**
>
> Regarding the additional probing stage in PKGE, we respectfully refer the reviewer to Table 14 in the Supplementary Material, which reports the time consumption for different training stages.
> Time Ratio: The results demonstrate that the probing phase accounts for a very small fraction of the overall training time.
> Data Strategy: In our practical implementation, we utilize only 1% of the data for probe training, while the remaining 99% is used for the subsequent MoE training. Therefore, compared to training on 100% of the data continuously, the specific design of our multi-stage pipeline ensures that the additional burden is **minimized** and does not create a bottleneck.

---

### Author Response · Authors · 2025-11-25
**Review and Reviewer-Author Discussion Summary**

Dear PCs, SACs, ACs, and Reviewers:

Thank you very much for your valuable contributions to our work. We thank all reviewers for their constructive feedback and for recognizing the value of LLaVA-CMoE. Below, we summarize the consensus on our contributions and detail how we have addressed specific concerns during the rebuttal period.

**1. Summary of Strengths**

We are encouraged that the reviewers recognized the innovations of our work:
- **Novelty**: Reviewer fo8y stated the paper is "well-written and clearly structured" and the modules "effectively address catastrophic forgetting and parameter efficiency". Reviewer xFvE noted that our work is "contributing new insights to Mixture of Experts architectures".
- **Effectiveness**: Reviewer xFvE highlighted that our performance is "significantly better than prior methods".
- **Well-motivated & Intuitive**: Reviewer xVKX mentioned "the motivation is well illustrated" and the workflow is "conceptually straightforward". Reviewer rkFQ called the solution "intuitive and simple" and "a step in the right direction".

**2. Key Improvements & Issues Resolved in Rebuttal**

We have conducted extensive additional experiments and analyses to address the reviewers' concerns as follows:
- **Comparison with Stronger Baselines**:
  - Concern: Baselines were considered out-of-date or insufficient (**Reviewer fo8y, Reviewer xVKX, Reviewer rkFQ**).
  - Our Solution: In the revision, we have supplemented comprehensive comparisons in Table 1 with multiple publicly available baseline models to better contextualize the performance of our approach. Besides, we also implemented and reproduced MoExtend [1]  within our unified codebase. Results show our method outperforms MoExtend on the majority of datasets (e.g., +1.58% on SQA) while adding significantly fewer experts (adaptive vs. fixed). We also addressed HiDe-LLaVA [2] (issued by Reviewer xVKX) by clarifying that its performance advantage stems from data leakage (pre-training on test set data), whereas our method respects strict continual learning boundaries.
- **Computational Cost & Inference Overhead**:
  - Concern: Request for FLOPs analysis and concerns about PTL inference overhead (**Reviewer fo8y, Reviewer xVKX**).
  - Our Solution: We provided a detailed FLOPs analysis. The added computation (1.184 GFLOPs) constitutes merely 0.4% of the visual encoder's cost, confirming the overhead is negligible.
- **Generalization & Zero-Shot Capabilities**:
  - Concern: Performance on domain shifts (e.g., medical) and unseen tasks (**Reviewer xFvE, Reviewer rkFQ**).
  - Our Solution: We conducted new experiments on PMC-VQA (Medical), where PGKE successfully triggered adaptive expansion (28 experts) to handle the distribution shift. We also added Zero-shot evaluation on VQAv2, achieving 48.32% accuracy without gradient updates, proving PTL effectively retrieves related historical knowledge.
- **Methodological Clarifications (Expert Diversity & Hyperparameters)**:
  - Concern: Expert initialization diversity, role of load balancing, and clarification on $N_s$ (**Reviewer xVKX, Reviewer rkFQ**).
  - Our Solution: We provided Cosine Similarity analysis showing that while new experts start as clones (warm start), they significantly diverge (similarity drops to ~0.6) after training. We also clarified that $N_s=1$ is the default hyperparameter and demonstrated the necessity of Load Balancing loss for rationalizing expert selection.

**3. Recognition of our revision from reviewers**

During the rebuttal period, we actively addressed all reviewers' concerns through substantial additional experiments and clarifications. **Reviewer xFvE** recognized our rebuttal and stated that **their concerns are resolved** ("Thanks for the clarification"), maintaining a positive score. **Reviewer xVKX** explicitly stated that "**most technical issues have been resolved**." Regarding their follow-up concern on recent baselines (HiDe-LLaVA [2]), we have provided a strong rebuttal identifying critical data leakage in the baseline method, reinforcing the validity of our results.

Importantly, the concerns raised by **Reviewer fo8y** and **Reviewer rkFQ** substantially overlapped with those raised by Reviewers xFvE and xVKX. As evidenced by Reviewers xFvE's and xVKX's satisfaction, the primary concerns of Reviewers fo8y and rkFQ, specifically regarding Baselines and Computational Cost, have been addressed. Therefore, we believe these shared concerns have been effectively resolved for all reviewers.

We hope this concise summary will assist the AC's work. Thanks again to all reviewers and chairs for their constructive feedback, which has significantly strengthened our paper.

**Reference:**

[1] Shanshan Zhong, Shanghua Gao, Zhongzhan Huang, Wushao Wen, Marinka Zitnik, and Pan Zhou. Moextend: Tuning new experts for modality and task extension. ACL'24

[2] HiDe-LLaVA: Hierarchical Decoupling for Continual Instruction Tuning of Multimodal Large Language Model. ACL'25

---

### Meta-Review · Area_Chair_HWkK · 2026-01-07

**Summary:**

The rebuttal adds additional analyses (e.g., FLOPs estimates and supplementary experiments) and provides further discussion to clarify several ambiguities. After the rebuttal, some reviewer concerns and questions regarding specific technical details appear to be addressed. However, the overall justification remains insufficiently convincing, as multiple issues related to the complexity of the proposed framework remain open or give rise to new concerns (xVKX). After carefully reviewing both the paper and the rebuttal, AC also shares concerns that overlap with those raised by the reviewers, as well as additional concerns from some different perspectives (detailed below in the reviewer concerns section). These issues collectively undermine a clear and well-substantiated explanation of where the performance gains originate and whether they justify the added methodological complexity. Oveerall, while the work is promising, the current submission is not yet ready for publication.

**Reviewer Concerns:**

Several reviewers (xVKX, rkFQ) raised concerns and questions regarding clarity and completeness in the description of the proposed method and experimental setup. Some of these issues were properly addressed in the rebuttal, including clarifications on the relationship between the reported “Immediate” results and oracle evaluations with ground-truth task IDs, as well as explanations of certain hyperparameter choices. However, concerns remain regarding the overall design and implementation details. In particular, after carefully examining the paper, AC seems not finding details about the specific version of the used LLaVA model (e.g., the training data), which is important for assessing fairness and reproducibility.

Reviewers xVKX, fo8y, and rkFQ also pointed out the insufficient coverage of baseline and related methods, especially those on similar data and experimental settings. While the rebuttal discusses some of these methods and provides partial comparisons and explanations (e.g., differences in data exposure), more rigorous and controlled experiments would be necessary for a publishable version of the paper. Although this issue is not fatal for rejection, it is important for establishing a clear experimental context and inforamtive analysis for readers.

Reviewer fo8y raised concerns about computational overhead, and xVKX asked about task locator.
While the rebuttal reports FLOPs estimates and provides qualitative explanations, several details remain unclear, limiting the ability to properly assess the computational justification of the proposed framework.

Upon closer checking of the details regarding the above questions, the AC also noted that an additional frozen LLaVA model is required to extract the feature F_end​ for task identification via a VAE. This raises two main concerns additionally. (1) It is unclear whether this reliance on a frozen full-size LLaVA implies that an additional forward pass through the entire LLaVA model is required at inference time. If so, it introduces substantial computational overhead. Based on the wording in the rebuttal, it appears that this cost may not have been fully accounted for in the reported FLOPs. While task level routing is commonly used, relying on an additional full size model is usually not an option. (2) The use of a VAE combined with z-score based metrics for expert allocation or MoE expansion has been explored in prior work and adopted by several related approaches. As such, the authors may need to more carefully check existing literatures and refine the positioning of their contribution.

Overall, although the rebuttal provides additional clarification, a number of reviewer concerns and open questions remain unresolved, either as explicitly raised by the reviewers or identified independently by the AC.

**Reviewer Scores:**

- fo8y. Likely maintain the original score 6. The rebuttal directly addresses the question on FLOPs and adds one newer baseline. However, broader issues regarding baseline strength and the efficiency of the overall framework remain.


- xFvE. Likely maintain 6. The authors responded to the reviewer’s questions. The reviewer stated to maintain the original score in response.

- xVKX. Likely to remain below the acceptance threshold, maybe maintain the original score of 4. While the rebuttal provides clarifications and additional analyses, it also surfaced new concerns followed up by the reviewer, regarding missing baselines and suggesting related works for comparison. The score might improve if the reviewer is fully convinced by the authors’ explanations.


- rkFQ. likely maintain the original 6. The rebuttal improves clarity on several design choices and adds a zero-shot evaluation. Concerns about complexity, clarity of the framework, and limited baseline remain and may influence further increase of the score.

---

### Decision · Program_Chairs · 2026-01-26

Reject